# EFFICIENT PLANNING IN A COMPACT LATENT ACTION SPACE

**Zhengyao Jiang**[1]* **Tianjun Zhang**[2] **Michael Janner**[2] **Yueying Li**[3]
**Tim Rocktäschel**[1] **Edward Grefenstette**[1,4] **Yuandong Tian**[5]
[1]University College London [2]University of California, Berkeley [3]Cornell University
[4]Cohere [5]Meta AI (FAIR)

## ABSTRACT

Planning-based reinforcement learning has shown strong performance in tasks in discrete and low-dimensional continuous action spaces. However, planning usually brings significant computational overhead for decision-making, and scaling such methods to high-dimensional action spaces remains challenging. To advance efficient planning for high-dimensional continuous control, we propose Trajectory Autoencoding Planner (TAP), which learns low-dimensional latent action codes with a state-conditional VQ-VAE. The decoder of the VQ-VAE thus serves as a novel dynamics model that takes latent actions and current state as input and reconstructs long-horizon trajectories. During inference time, given a starting state, TAP searches over discrete latent actions to find trajectories that have both high probability under the training distribution and high predicted cumulative reward. Empirical evaluation in the offline RL setting demonstrates low decision latency which is indifferent to the growing raw action dimensionality. For Adroit robotic hand manipulation tasks with high-dimensional continuous action space, TAP surpasses existing model-based methods by a large margin and also beats strong model-free actor-critic baselines.

## 1 INTRODUCTION

Planning-based reinforcement learning (RL) methods have shown strong performance on board games (Silver et al., 2018; Schrittwieser et al., 2020), video games (Schrittwieser et al., 2020; Ye et al., 2021) and low-dimensional continuous control (Janner et al., 2021). Planning conventionally occurs in the raw action space of the Markov Decision Process (MDP), by rolling-out future trajectories with a dynamics model of the environment, which is either predefined or learned.

While such a planning procedure is intuitive, planning in raw action space can be inefficient and inflexible. Firstly, the optimal plan in a high-dimensional raw action space can be difficult to find. Even if the optimizer is powerful enough to find the optimal plan, it is still difficult to make sure the learned model is accurate in the whole raw action space. In such cases, the planner can exploit the weakness of the model and lead to over-optimistic planning. Secondly, planning in raw action space means the planning procedure is tied to the temporal structure of the environment. However, human planning is much more flexible, for example, humans can introduce temporal abstractions and plan with high-level actions; humans can plan backwards from the goal; the plan can also start from a high-level outline and get refined step-by-step. The limitations of raw action space planning cause slow decision speeds, which hamper adoption in real-time control.

In this paper, we propose the Trajectory Autoencoding Planner (TAP), which learns a latent action space and latent-action model from offline data. A latent-action model takes state $s_1$ and latent actions $z$ as input and predicts a segment of future trajectories $\tau = (a_1, r_1, R_1, s_2, a_2, r_2, R_2, ...)$. This latent action space can be much smaller than the raw action space since it only captures plausible trajectories in the dataset, preventing out-of-distribution actions. Furthermore, the latent action decouples the planning from the original temporal structure of MDP. This enables the model, for example, to predict multiple steps of future trajectories with a single latent action.

---

*Correspond to z.jiang@cs.ucl.ac.uk. The webpage is at: `sites.google.com/view/latentplan`. Source code is available at: `github.com/ZhengyaoJiang/latentplan`.

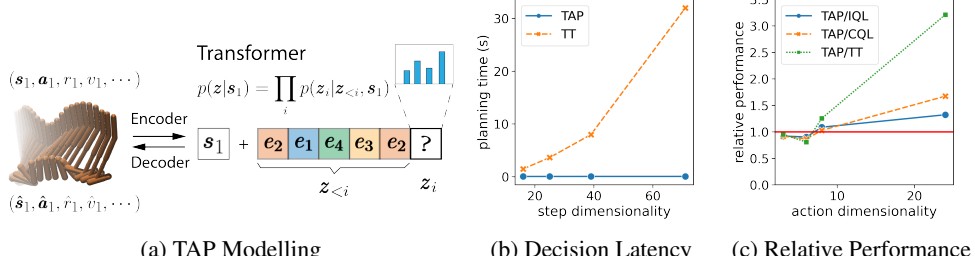

(a) TAP Modelling        (b) Decision Latency        (c) Relative Performance

Figure 1: (a) gives an overview of TAP modelling, where blocks represent the latent actions. (b) shows decision time growth with the dimensionality $D$. Tests are done on a single GPU. The number of planning steps for (b) is 15 and both models apply a beam search with a beam width of 64 and expansion factor of 4. (c) shows the relative performance between TAP and baselines when dealing with tasks with increasing raw action dimensionalities.

The model and latent action space are learned in an unsupervised manner. Given the current state, the encoder of TAP learned to map the trajectories to a sequence of discrete latent codes (and the decoder maps it back), using a state-conditioned Vector Quantised Variational AutoEncoder (VQ-VAE). As illustrated in Figure 1(a), the distribution of these latent codes is then modelled autoregressively with a Transformer, again conditioned on the current state. During inference, instead of sampling the actions and next state sequentially, TAP samples latent codes, reconstructs the trajectory via the decoder, and executes the first action in the trajectory with the highest objective score. These latent codes of the VQ-VAE are thus latent actions and the state-conditional decoder acts as a latent-action dynamics model. In practice, TAP uses a single discrete latent variable to model multiple ($L = 3$) steps of transitions, creating a compact latent action space for downstream planning. Planning in this compact action space reduces the decision latency and makes high-dimensional planning easier. In addition, reconstructing the entire trajectories after all latent codes are sampled also helps alleviate the compound errors of step-by-step rollouts.

We evaluate TAP extensively in the offline RL setting. Our results on low-dimensional locomotion control tasks show that TAP enjoys competitive performance as strong model-based, model-free actor-critic, and sequence modelling baselines. On tasks with higher dimensionality, TAP not only surpasses model-based methods like MOPO (Yu et al., 2020) Trajectory Transformer(TT) (Janner et al., 2021) but also significantly outperforms strong model-free ones (e.g., CQL (Kumar et al., 2020)and IQL (Kostrikov et al., 2022)). In Figure 1(c), we show how the relative performance between TAP and baselines changes according to the dimensionality of the action space. One can see that the advantage of TAP starts to pronounce when the dimensionality of raw actions grows and the margin becomes large for high dimensionality, especially when compared to the model-based method TT. This can be explained by the innate difficulty of policy optimization in a high-dimensional raw action space, which is avoided by TAP since its planning happens in a low-dimensional discrete latent space. At the same time, the sampling and planning of TAP are significantly faster than prior work that also uses Transformer as a dynamics model: the decision time of TAP meets the requirement of deployment on a real robot (20Hz) (Reed et al., 2022), while TT is much slower and the latency grows along the state-action dimensionality in Figure 1(b).

## 2 BACKGROUND

### 2.1 VECTOR QUANTISED-VARIATIONAL AUTOENCODER

The Vector Quantised Variational Autoencoder (VQ-VAE) (van den Oord et al., 2017) is a generative model designed based on the Variational Autoencoder (VAE). There are three components in VQ-VAEs: 1) an encoder network mapping inputs to a collection of discrete latent codes; 2) a decoder that maps latent codes to reconstructed inputs; 3) a learned prior distribution over latent variables. With the prior and the decoder, we can draw samples from the VQ-VAE.

In order to represent the input with discrete latent variables, VQ-VAE employs an approach inspired by vector quantization (VQ). The encoder outputs continuous vectors $\boldsymbol{x}_i$ which are not input to the decoder directly but used to query a vector in a codebook $\mathbb{R}^{K \times D}$ with $K$ embedding vectors $\boldsymbol{e}_k \in \mathbb{R}^D, k \in 1, 2, 3, ..., K$. The query is done by simply finding the nearest neighbour from $K$ embedding vectors, and the resultant vectors

$$\boldsymbol{z}_i = \boldsymbol{e}_k, \text{where } k = \text{argmin}_j \, ||\boldsymbol{x}_i - \boldsymbol{e}_j||_2 \tag{1}$$

are input to the decoder. The autoencoder is trained by jointly minimizing the reconstruction error and the distances $||\boldsymbol{x}_i - \boldsymbol{e}_k||_2$ and $||\boldsymbol{z}_i - \boldsymbol{e}_k||_2$. The gradients are copied directly through the bottleneck during backpropagation. A PixelCNN (van den Oord et al., 2016) is used to model the prior distribution of the latent codes autoregressively.

## 3 METHOD

To enable flexible planning in learned latent action space, we propose Trajectory Autoencoding Planner (TAP) based on the VQ-VAE. In this section, we will describe the latent action model framework and provide the model architecture details for TAP. Then we elaborate on how to plan with this latent-action model.

### 3.1 LATENT-ACTION MODEL AND PLANNING

Consider the following trajectory $\tau$ of length $T$, sampled from an MDP with a fixed stochastic behaviour policy, consisting of a sequence of states, actions, rewards and return-to-go $R_t = \sum_{i=t} \gamma^{i-t} r_i$ as proxies for future rewards:

$$\tau = (\boldsymbol{s}_1, \boldsymbol{a}_1, r_1, R_1, \boldsymbol{s}_2, \boldsymbol{a_2}, r_2, R_2, \ldots, \boldsymbol{s}_T, \boldsymbol{a}_T, r_T, R_T). \tag{2}$$

We model the conditional distribution of the trajectory $p(\tau|\boldsymbol{s}, z)$ with a sequence of latent variables $z = (\boldsymbol{z}_1, \ldots, \boldsymbol{z}_M)$. Assume the state and latent variables $(\boldsymbol{s}, z)$ can be deterministically mapped to a trajectory $\tau$ so that $p(\tau|\boldsymbol{s}, z) = \mathbb{1}(\tau = h(\boldsymbol{s}, z))p(z|\boldsymbol{s})$. We refer to $z$ as *latent actions* and $p(z|\boldsymbol{s})$ as the *latent policy*. The mapping $h(\boldsymbol{s}, z)$ from state and latent actions $(\boldsymbol{s}, z)$ to a trajectory $\tau$ is thus a *latent-action model*. In a deterministic MDP, the trajectory for an arbitrary $h(\boldsymbol{s}, z)$ with $p(z|\boldsymbol{s}) > 0$ will be an executable plan, namely, the trajectory can be recovered by following the action sequences, starting from state $\boldsymbol{s}$. Therefore, we can optimize latent actions $z$ in order to find an optimal plan.

Planning in the latent action space has two advantages compared to planning in the raw action space. Firstly, the latent-action model only captures possible actions in the support of the behaviour policy, allowing the latent action space to potentially be much smaller than the raw action space. For example, considering the policy is a mixture of $X$ policies, then the latent action space can be a discrete space with only $X$ actions, no matter how high-dimensional the raw action space is. In addition, only allowing the in-distribution actions prevents the planner from exploiting the weakness of the model by querying the actions with high uncertainty whose values are most susceptible to overestimation. Secondly, the latent-action model allows the decoupling of the temporal structure between planning and modelling. When planning in the raw action space, the time resolution of planning must be the same as the predicted trajectories. In contrast, planning in the latent action space can be much more flexible, as a latent action does not have to be tied to a particular step of the transition. This property allows the length of the latent action sequence $M$ to be smaller than the planning horizon $T$, leading to more efficient planning.

### 3.2 LEARNING A LATENT-ACTION MODEL WITH VQ-VAES

A discrete action space makes dealing with the full distribution of actions easier and also allows a spectrum of advanced planning algorithms to be applied Silver et al. (2018); Tillmann et al. (1997). To learn a compact discrete latent action space, we propose Trajectory Autoencoding Planner (TAP) to model the trajectories with a state-conditioned VQ-VAE. We treat $\boldsymbol{x}_t := (\boldsymbol{s}_t, \boldsymbol{a}_t, r_t, R_t)$ as a single token for the Transformer encoder and decoder. Both the autoencoder and the prior over latent variables are conditioned on the first state $\boldsymbol{s}_1$ of the trajectory.

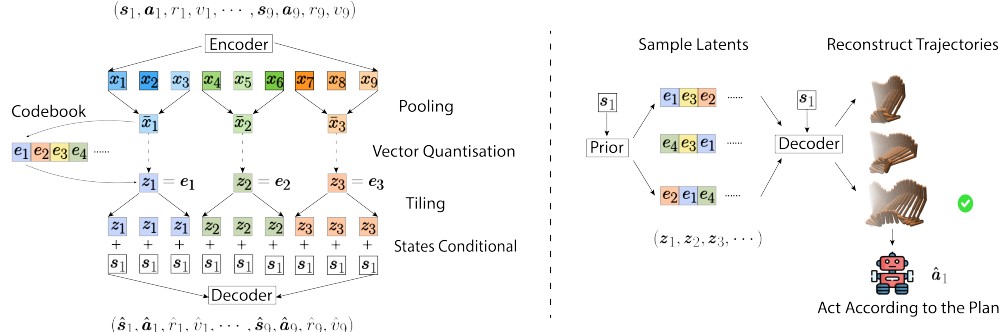

Figure 2: Illustration of the training and test time inference process of TAP. The left-hand side shows the training process, highlighting the design of the bottleneck. The right-hand side figure shows how we generate plans during the test time, with vanilla sampling.

**Encoder $g$ and decoder $h$.** For the encoder, token $x_t$ is processed by a causal Transformer, leading to $T$ feature vectors. We then apply a 1-dimensional max pooling with both kernel size and stride of $L$, followed by a linear layer. This results in $M = T/L$ vectors $(\bar{\boldsymbol{x}}_1, \bar{\boldsymbol{x}}_2, ..., \bar{\boldsymbol{x}}_M)$, corresponding to the number of discrete latent variables. After vector quantization in Equation (1), we get embedding vectors for the latent variables $(\boldsymbol{z}_1, \boldsymbol{z}_2, ..., \boldsymbol{z}_M)$.

For the decoder, the latent variables are then tiled $L$ times to match the number of the input/output tokens $T$. For $L = 3$, this is written as $\text{tile}(\boldsymbol{z}_1, \boldsymbol{z}_2, ..., \boldsymbol{z}_{T/3}) = (\boldsymbol{z}_1, \boldsymbol{z}_1, \boldsymbol{z}_1, \boldsymbol{z}_2, \boldsymbol{z}_2, \boldsymbol{z}_2, ..., \boldsymbol{z}_{T/3}, \boldsymbol{z}_{T/3}, \boldsymbol{z}_{T/3})$. We concatenate the state and embedding vector for the codes and apply a linear projection to provide state information to the decoder. After adding the positional embedding, the decoder then reconstructs the trajectory $\hat{\tau} := (\hat{\boldsymbol{x}}_1, \hat{\boldsymbol{x}}_2, ..., \hat{\boldsymbol{x}}_T)$, where $\hat{\boldsymbol{x}}_t := (\hat{\boldsymbol{s}}_t, \hat{\boldsymbol{a}}_t, \hat{r}_t, \hat{R}_t)$. Finally, to train the encoder/decoder, the reconstruction loss is the mean squared error over the input trajectories $\{\tau\}$ and reconstructed ones $\{\hat{\tau}\}$.

**Prior distribution of latent codes**. While the derivation of VQ-VAE assumes a uniform prior over latents, in practice, learning a parameterised prior over latents will usually lead to better sample quality. Similarly, TAP learns a prior policy $p(\boldsymbol{z}|\boldsymbol{s}_1)$ to inform sampling and planning. TAP uses a Transformer to autoregressively model the distribution of the latent action codes given the initial state $\boldsymbol{s}_1$. The state information is also blended into the architecture by the concatenation of token embeddings. We denote the autoregressive prior policy as $p(\boldsymbol{z}_t|\boldsymbol{z}_{<t}, \boldsymbol{s}_1) = p(\boldsymbol{z}_t|\boldsymbol{s}_1, \boldsymbol{z}_1, \boldsymbol{z}_2, ..., \boldsymbol{z}_{t-1})$.

### 3.3 PLANNING IN THE DISCRETE LATENT ACTION SPACE

We now describe how to plan with the learned TAP model. This includes how to evaluate trajectories and optimize latent actions.

**Evaluation Criterion**. Given the initial state and latent actions $(\boldsymbol{s}_1, z)$, we obtain a sample trajectory $\hat{\tau} = (\hat{\boldsymbol{x}}_1, ..., \hat{\boldsymbol{x}}_T)$ from decoder $h$, where $\hat{\boldsymbol{x}}_t := (\hat{\boldsymbol{s}}_t, \hat{\boldsymbol{a}}_t, \hat{r}_t, \hat{R}_t)$. We score $\tau$ with the following function $g$:

$$g(\boldsymbol{s}_1, \boldsymbol{z}_1, \boldsymbol{z}_2, ..., \boldsymbol{z}_M) = \sum_t \gamma^t \hat{r}_t + \gamma^T \hat{R}_T + \alpha \ln\left(\min(p(\boldsymbol{z}_1, \boldsymbol{z}_2, ..., \boldsymbol{z}_M|\boldsymbol{s}_1), \beta^M)\right) \tag{3}$$

The first part (in red) of the score function is the predicted return-to-go following the action sequences in the decoded trajectory $\hat{\tau}$. The second part (in blue) is a term that penalizes out-of-distribution plans, where the prior distribution $p$ gives low probability. The hyperparameter $\alpha$ is set to be larger than the maximum of the discounted returns to select a plausible trajectory when the conditional probability of latent action sequences is lower than the threshold $\beta^M$. When the probability is higher than the threshold, the objective will then encourage choosing a high-return plan.

**Vanilla Sampling**  The most straightforward way to search in the latent space of TAP is to sample according to the prior distribution and select the trajectory that gives the best return. Conditioned on current state $\boldsymbol{s}_1$, we can sample $N$ latent variable sequences autoregressively according to the prior model: $(\boldsymbol{z}_1, \boldsymbol{z}_2, ..., \boldsymbol{z}_M)_n \sim p(\boldsymbol{z}_1, \boldsymbol{z}_2, ..., \boldsymbol{z}_M | \boldsymbol{s}_1)$ where $n \in 1, 2, 3, ..., N$. These latent variable sequences will then be turned into trajectories $\tau_n$ so that we can select the optimal one according to the objective function Equation (3). This naïve approach works well when the overall length of the trajectory is small since the discrete latent space is significantly smaller than the raw action space. However, its performance quickly degenerates with long planning horizons.

**Beam Search in the Latent Space**  We use causal Transformers for encoder, decoder and the prior policy, preventing the decoding depending on the future latent actions. This allows us to partially decode the trajectory of length $NL$ with first $N$ latent variables. Therefore we can apply a planning algorithm like *beam search* for more efficient optimization by only expanding promising branches of the search tree. TAP beam search always keeps $B$ best partial latent code sequences, and samples $E$ new latent actions conditioned on the partial codes. Here $B$ is the beam width and $E$ is the expansion factor. The pseudocode of the TAP beam search is shown in Algorithm 1 in the Appendix. In practice, we find TAP with beam search is usually computationally more efficient because it can find better trajectories with fewer queries to the prior model and decoder.

## 4 Experiments

The empirical evaluation of TAP consists of three sets of tasks from D4RL (Fu et al., 2020): gym locomotion control, AntMaze, and Adroit. We compare TAP to a number of prior offline RL algorithms, including both model-free actor-critic methods (Kumar et al., 2020; Kostrikov et al., 2022) and model-based approaches (Kidambi et al., 2020; Yu et al., 2020; Lu et al., 2022). Our work is conceptually the most related to the Trajectory Transformer (TT; Janner et al. 2021), a model-based planning method that predicts and plans in the raw state and action space, so this baseline serves as our main point of comparison. Gym locomotion tasks serve as a proof of concept in the low-dimensional domain to test if TAP can accurately reconstruct trajectories for use in decision-making and control. We then test TAP on Adroit, which has a high state and action dimensionality that causes TT to struggle because of its autoregressive modelling of long token sequences. Finally, we also test TAP on AntMaze, a sparse-reward continuous-control problem. TAP achieves similar performance as TT on AntMaze, surpassing model-free methods, where the details can be found in Appendix E.

### 4.1 Setup for Trajectory Autoencoding Planner

We keep most of the design decisions the same as in TT in order to show the advantage of the latent-action model. Namely, (1) no bootstrapped value estimation is used; (2) the trajectory is treated as a sequence and the latent structure of the model does not follow the structure of the MDP; (3) beam search is used for planning; and (4) the model architectures for the encoder, decoder, and prior policy are Transformers.

As for the TAP-specific hyperparameters: we set the number of the steps associated with each latent variable to be $L = 3$ and each latent variable has $K = 512$ candidate values. The planning horizon in the raw action space is 15 for gym locomotion tasks and 24 for Adroit tasks. As such, the planning horizon in the latent space will be reduced to 5 or 8. Other hyperparameters including architectures can be found in the Appendix. We test each task with 5 training seeds, each evaluated for 20 episodes. Following the evaluation protocol of TT and IQL, we use the v2 version of the datasets for locomotion control and v0 for the other tasks.

### 4.2 Results

On low-dimensional gym locomotion control tasks, all the model-based methods show comparable performance to model-free ones. However, the performance of model-based baselines on high-dimensional adroit tasks is much worse than the low-dimensional case, showing inferior scalability with respect to state and action dimensions. In contrast, TAP shows consistently strong performance

among low-dimensional and high-dimensional tasks and surpasses other model-based methods with a large margin on Adroit.

To show how the relative performance between TAP and baselines vary with action dimension, we average the relative performance on tasks of the same action dimensionality and plotted them in Figure 1(c). [1] The x-axis is the action dimensionality of the tasks and the horizontal red line shows the relative performance of 1, meaning two methods achieve the same score. We can see that TAP scales better in terms of decision latency compared to TT. Also, TAP shows better performance for tasks with higher action dimensionality, compared to both TT and strong model-free actor-critic methods (CQL/IQL). This can be explained by the increasing difficulty of policy optimization in larger action space due to the curse of dimensionality. In contrast, TAP does the optimization in a compact latent space with a handful of discrete latent variables.

Table 1: Locomotion control results. Numbers are normalised scores following the protocol of Fu et al. (2020).

| Type | | Model-free | | | | Model-based | | |
|---|---|---|---|---|---|---|---|---|
| **Dataset** | **Environment** | **BC** | **CQL** | **IQL** | **DT** | **MOReL** | **TT** | **TAP (Ours)** |
| Medium-Expert | HalfCheetah | 59.9 | 91.6 | 86.7 | 86.8 | 53.3 | **95.0** | 91.8 ±0.8 |
| Medium-Expert | Hopper | 79.6 | 105.4 | 91.5 | 107.6 | 108.7 | **110.0** | 105.5 ±1.7 |
| Medium-Expert | Walker2d | 36.6 | 108.8 | **109.6** | 108.1 | 95.6 | 101.9 | 107.4 ±0.9 |
| Medium-Expert | Ant | 114.2 | 115.8 | 125.6 | 122.3 | − | 116.1 | **128.8** ±2.4 |
| Medium | HalfCheetah | 43.1 | 44.4 | **47.4** | 42.6 | 42.1 | 46.9 | 45.0 ±0.1 |
| Medium | Hopper | 63.9 | 58.0 | 66.3 | 67.6 | **95.4** | 61.1 | 63.4 ±1.4 |
| Medium | Walker2d | 77.3 | 72.5 | 78.3 | 74.0 | 77.8 | **79.0** | 64.9 ±2.1 |
| Medium | Ant | 92.1 | 90.5 | **102.3** | 94.2 | − | 83.1 | 92.0 ±2.4 |
| Medium-Replay | HalfCheetah | 4.3 | **45.5** | 44.2 | 36.6 | 40.2 | 41.9 | 40.8 ±0.6 |
| Medium-Replay | Hopper | 27.6 | **95.0** | 94.7 | 82.7 | 93.6 | 91.5 | 87.3 ±2.3 |
| Medium-Replay | Walker2d | 36.9 | 77.2 | 73.9 | 66.6 | 49.8 | **82.6** | 66.8 ±3.1 |
| Medium-Replay | Ant | 89.2 | 93.9 | 88.8 | 88.7 | − | 77.0 | **96.7** ±1.4 |
| **Mean** | | 60.4 | 83.2 | 84.1 | 81.5 | − | 82.2 | 82.5 |

Table 2: Adroit robotic hand control results. These tasks have high action dimensionality (24 degrees of freedom).

| Type | | Model-free | | | Model-based | | | |
|---|---|---|---|---|---|---|---|---|
| **Dataset** | **Environment** | **BC** | **CQL** | **IQL** | **MOPO** | **Opt-MOPO** | **TT** | **TAP (Ours)** |
| Human | Pen | 34.4 | 37.5 | 71.5 | 6.2 | 19.0 | 36.4 | **76.5** ±8.5 |
| Human | Hammer | 1.5 | **4.4** | 1.4 | 0.2 | 0.5 | 0.8 | 1.4 ±0.1 |
| Human | Door | 0.5 | **9.9** | 4.3 | − | − | 0.1 | 8.8 ±1.1 |
| Human | Relocate | 0.0 | 0.2 | 0.1 | − | − | 0.0 | 0.2 ±0.1 |
| Cloned | Pen | 56.9 | 39.2 | 37.3 | 6.2 | 23.0 | 11.4 | **57.4** ±8.7 |
| Cloned | Hammer | 0.8 | 2.1 | 2.1 | 0.2 | **5.2** | 0.5 | 1.2 ±0.1 |
| Cloned | Door | −0.1 | 0.4 | 1.6 | − | − | −0.1 | **11.7** ±1.5 |
| Cloned | Relocate | −0.1 | −0.1 | −0.2 | − | − | −0.1 | −0.2 ±0.0 |
| Expert | Pen | 85.1 | 107.0 | − | 15.1 | 50.6 | 72.0 | **127.4** ±7.7 |
| Expert | Hammer | 125.6 | 86.7 | − | 6.2 | 23.3 | 15.5 | **127.6** ±1.7 |
| Expert | Door | 34.9 | 101.5 | − | − | − | 94.1 | **104.8** ±0.8 |
| Expert | Relocate | 101.3 | 95.0 | − | − | − | 10.3 | **105.8** ±2.7 |
| **Mean (without Expert)** | | 11.7 | 11.7 | 14.8 | − | − | 6.1 | **19.6** |
| **Mean (all settings)** | | 36.7 | 40.3 | − | − | − | 20.1 | **51.9** |

## 4.3 DECISION LATENCY

Transformer-based trajectory generative models (such as TT and Gato (Reed et al., 2022)) treat each dimension of the state and action as an individual token. Denoting $S$ as the dimensionality of the

---

[1] We did not include expert datasets for Adroit when computing average since the common protocol for gym locomotion evaluation also doesn't include expert datasets and the IQL paper didn't report these scores.

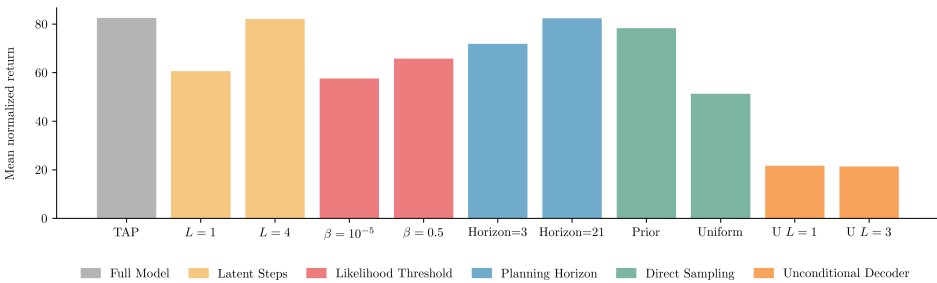

Figure 3: Results of ablation studies, where the height of the bar is the mean normalised scores on gym locomotion control tasks.

state space and $A$ as that of the action space, TT requires $D = S + A + 2$ tokens to model a step of a trajectory. Since the time complexity of the Transformer is $O(N^2)$ for a sequence of length $N$, the computational cost of TAP is significantly lower especially when the state-action dimensionality is high. Specifically, for a trajectory with $T$ steps, TAP uses $T$ tokens (in autoencoders) and TT uses $TD$ tokens. Therefore, the inference cost of TAP is $O(T^2)$ rather than $O(D^2T^2)$ of TT, leading to more efficient training and planning.

Besides the computational complexity of the forward pass and sampling, we also tested how exactly the computational costs and decision latency of TT and TAP grow along with the state-action dimensionality of the tasks. Tests are done on a platform with an i5 12900K CPU and a single RTX3090 GPU. We fix the planning horizon to be 15 (the default planning horizon for TT) and test the decision latency on hopper ($D = 14$), halfcheetah ($D = 23$), ant ($D = 37$) and Adroit-pen ($D = 71$) tasks. Even for the task with the lowest dimensionality, TT needs 1.5 seconds to make a decision and the latency grows to 32 seconds when dealing with adroit. On the other hand, TAP manages to make a decision within 0.05 seconds and meets the real-world robotics deployment criterion proposed by Reed et al. (2022). Moreover, the latency remains nearly constant for different dimensionalities, making it computationally feasible to be applied to high-dimensional control tasks.

## 4.4 ANALYSIS

TAP is a novel model-based RL method which involves several designs that can potentially be transferred to other model-based or offline RL methods. In order to provide more insights into these design choices, we provide analyses here, together with ablations of common hyper-parameters like planning horizon. In Figure 3, we showed a summary of the results of ablation studies on gym locomotion tasks. The full results for more hyper-parameters and scores for each individual task can be found in the Appendix.

**Latent Steps** A prominent feature of TAP is that planning happens in a latent action space with temporal abstraction. Whenever a single latent code is sampled, $L$ steps of transitions that continue the previous trajectory can be decoded. This design increases the efficiency of planning since the number of steps of unrolling is reduced to $\frac{1}{L}$; therefore, the search space is exponentially reduced in size. However, it is unclear how such a design affects the decision-making performance of the method. We thus tested TAP with different latent steps $L$, namely, the number of steps associated with a latent variable. As shown in the yellow bars in Figure 3, reducing the latent step $L$ to 1 significantly undermines the performance of TAP. We hypothesize that the performance drop is related to the overfitting of the VQ-VAE since we observe the reduced latent step leads to a higher estimated return and also a higher prediction error; see Appendix H for more details. On the other hand, we found in Appendix Table 8 that the optimal latent steps vary across tasks. This indicates a fixed latent step may also be suboptimal. We believe this shows a direction for future work to further decouple the temporal structure between latent actions and decoded trajectories; see Appendix D for further discussion.

**OOD Penalty** By varying the threshold for probability clipping ($\beta$), we also tested the benefits of having both an OOD penalty and return estimations in the objective function Equation (3). When

$\beta = 10^{-5}$, the estimated return term dominates the optimization objective for the reward scale of the tasks considered. When $\beta = 0.5$, the OOD penalty is always activated and encourages the agent to choose the most likely trajectory, namely, imitating the behaviour policy. In these two cases, the performance of the TAP agent drops 30.2% and 20.2%, respectively. This experiment demonstrates that solely optimizing the return or prior probability leads to suboptimal performance, so both ingredients are necessary for the objective function. Nevertheless, besides these extreme values, the performance of TAP is robust to a wide range of choices of $\beta$, ranging from 0.002 to 0.1, as shown in Table 4.

**Planning Horizon**  As shown in the blue bars in Figure 3, TAP is not very sensitive to the planning horizon, and achieves a mean score of 71.9 (-12.9%) by simply expanding a single latent variable, thus doing 3 steps of planning. This conclusion might be bound to the tasks, as dense-reward locomotion control may require less long-horizon reasoning than more complex decision-making problems. An ablation (Hamrick et al., 2021) of MuZero also shows that test-time planning is not as helpful in more reactive games like Atari compared to board games.

**Beam Search**  We also investigate whether structured decoding in the form of beam search is helpful, compared to an alternative strategy of simply autoregressive sampling from the prior policy to generate action proposals. We test the TAP agent with this form of random shooting by sampling 2048 trajectories from the policy prior. As shown in the green bar with the "Prior" label in Figure 3, beam search still performs slightly better. Moreover, its decision latency is lower because the beam width (64) is much smaller than the number of samples needed for direct sampling (2048). On the other hand, the fact that even direct sampling can generate decent plans for TAP shows that the latent space for TAP is compact and can be used for efficient planning.

**Sampling from a uniform action prior**  When TAP performs beam search or direct sampling, the latent action codes are sampled from the learned prior policy rather than the uniform distribution. Such a design is straightforward for a VQ-VAE since the objective, in that case, is to sample from the dataset distribution. However, in the case of RL, the aim is different since we would like to find an optimal trajectory within the support of the data. We therefore also test direct sampling where the latent codes are uniformly sampled. The performance of this approach is illustrated as the green bar with a "Uniform" label in Figure 3. Sampling according to the uniform distribution largely damages (-37.9%) the performance of the agent, even with an OOD penalty. To further investigate the role of sampling from the prior, in Appendix I in we visualize the distribution of trajectories modelled by TAP in two locomotion tasks.

**Unconditional Decoder**  One of the key design choices of TAP is conditioning the decoder on the initial state of a trajectory. This allows us to model the (conditional) distribution of trajectories with a very small number of latent variables in order to construct a compact action space for downstream planning. While it might be obvious that representing multiple steps of high-dimensional states and actions with a single latent variable is difficult, we present an empirical investigation of the performance of TAP with the unconditional decoder. The orange bars in Figure 3 show that without the first state as input to the decoder, the performance of TAP drops drastically. This is because, given the same number of latent variables, the reconstruction accuracy of the unconditional decoder is much lower. Slightly increasing the number of latent variables from $L = 3$ to $L = 1$ also does not alleviate the performance drop. Details about this ablation can be found in the Appendix Table 10.

## 5    RELATED WORK

TAP fits into a line of work on model-based RL method  (Sutton, 1990; Silver et al., 2008; Fairbank, 2008; Deisenroth & Rasmussen, 2011; Lampe & Riedmiller, 2014; Heess et al., 2015; Wang & Ba, 2020; Schrittwieser et al., 2020; Amos et al., 2021; Hafner et al., 2021) because it makes decisions by predicting into the future. In such approaches, prediction often occurs in the original state space of an MDP, meaning that models take a current state and action as input and return distribution over future states and rewards. TAP is different from these conventional model-based methods since the TAP takes latent actions and the current state as inputs, returning a whole trajectory. These latent variables act as a learned latent action space which is compact and allows efficient planning.

Hafner et al. (2019) and Ozair et al. (2021) proposed to learn a latent state space and dynamics function in the latent space. However, in their cases, the action space of the planner is still the same as the raw MDP and the unrolling of the plan is still tied to the original temporal structure of the environment. Wang et al. (2020); Yang et al. (2021) proposed to learn representations of actions on-the-fly for black-box optimization and path planning setting. While related at a high level, these works operate on environment dynamics that are assumed to be known in advance. TAP extends this idea to the RL setting where the true dynamics of the environment are unknown, requiring a dynamics model and latent action representation space to be learned jointly.

TAP follows a recent line of work that treats RL as a sequence modelling problem (Chen et al., 2021; Janner et al., 2021; Zheng et al., 2022). Such works have used GPT-2 (Radford et al., 2019) style Transformers (Vaswani et al., 2017) to model the whole trajectory of states, actions, rewards and values, and turn the prediction ability into a policy. DT applies an Upside Down RL (Schmidhuber, 2019) approach to predict actions conditioned on rewards. TT applies planning in order to find the best trajectory that gives the highest return. The strong sequence modelling power of the Transformer enables TT to generate long-horizon plans with high accuracy. TAP provides an efficient planning solution for TT and all other discrete space planning algorithms to efficiently plan in complex action spaces.

The concept of learning a representation of actions and doing RL in this latent action space (or skill embedding space) has been explored in model-free methods (Dadashi et al., 2022; Allshire et al., 2021; Zhou et al., 2020; Merel et al., 2019; Peng et al., 2022). Unlike TAP, where latent action space is to enable efficient and robust planning, the motivations for learning a latent action space for model-free methods are varied but the key point is to provide policy constraints. For example, Merel et al. (2019) and Peng et al. (2022) apply this idea to humanoid control to make sure the learned policies are human-like. Zhou et al. (2020) used latent actions to prevent out-fo-distribution (OOD) actions in offline RL setting. Dadashi et al. (2022) proposes a discrete latent action space to make methods designed for discrete action space to be applied to the continuous case. In the literature of teleportation, Losey et al. (2022); Karamcheti et al. (2021) embed the robot's high-dimensional actions into low-dimensional and human-controllable latent actions.

In this paper, TAP is tested in an offline RL setting (Ernst et al., 2005) where we are not allowed to use online experience to improve our policy. A major challenge of offline RL is to prevent out-of-distribution (OOD) actions to be chosen by the learned policy, so that the inaccuracy of the value function and the model will not be exploited. Conservatism has been proposed as a solution to this issue (Kumar et al., 2020; Fujimoto & Gu, 2021; Lu et al., 2022; Kostrikov et al., 2022; Kidambi et al., 2020). TAP can naturally alleviate the OOD actions because it models trajectories generated by behaviour policy and decoded policy is therefore in-distribution.

In the literature of hierarchical RL, a conceptually relevant work to ours is (Co-Reyes et al., 2018), which uses a VAE to project the state sequences into continuous latent variables and conditions policies on not only the current state but also the latent variable. Similar ideas also appear in goal-oriented imitation learning. Lynch et al. (2019) proposes to condition a goal-reaching imitation policy on a latent variable of trajectories to avoid averaging out multiple modes of the distribution of trajectories. Mandlekar et al. (2020) use a VAE to model the distribution of reachable states and filter the sampled states according to a value function, then feed the goal to a downstream goal-reaching policy. Since the high-level latent actions are discrete, our work is also related to the (conceptual) options framework (Sutton et al., 1999; Stolle & Precup, 2002; Bacon et al., 2017) since both the latent codes of TAP and options introduce a mechanism for temporal abstraction.

## 6 LIMITATIONS

One of the limitations of TAP is that it does not distinguish between uncertainty stemming from a stochastic behaviour policy and uncertainty from the environment dynamics. Our empirical results on continuous control with deterministic dynamics suggest that TAP can handle epistemic uncertainty, potentially thanks to the OOD penalty in the objective function. However, it is unclear if TAP can handle tasks with stochastic dynamics without any modifications. Another problem for the latent-action model is the loss of fine-grained control over the generated trajectories. For example, it is nontrivial for TAP to predict future trajectories conditioned on a sequence of actions in the raw action space, which is straightforward to do with conventional dynamics models.

**Acknowledgements** The work is supported by UCL Digital Inovation Center and AWS. We thank to Minqi Jiang, Robert Kirk, Akbir Khan and Yicheng Luo for insightful discussions.

## 7 REPRODUCIBILITY STATEMENT

The code to reproduce all the experiment results are available in the supplementary materials. README includes guides to setup and environment and commands to run the experiments. The hyperparameters used in our experiments are in the default config file which will be loaded automatically. Experiments for all environments can be run on a single GPU. Besides codes, pseudo-code in Appendix Algorithm 1 and hyperparameters described in Appendix Table 7 can also be helpful to reproduce the results if readers are interested in implementing by themselves.

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

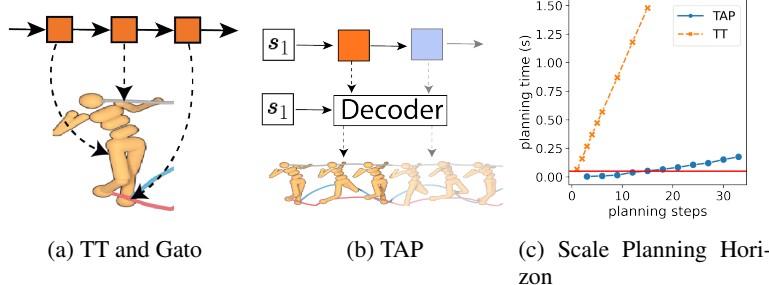

|                    |                |                                   |
| ------------------ | -------------- | --------------------------------- |
| (a) TT and Gato    | (b) TAP        | (c) Scale Planning Horizon        |

Figure 4: (a) Illustration of dimension-wise autoregressive modelling used by Trajectory Transformer. Blocks represent discretised state/action dimensions or discrete latent variables. (b) shows TAP style modelling. (c) Horizon Scalability, tested for a low-dimensional task, hopper.

---

**Algorithm 1** TAP Beam Search

---

**Input:** Current state $s$, sequence length $T$, beam width $B$, expansion factor $E$, prior model $p$ and decoding function $h$

1: Sample initial context latents $Z_1 = \{z_1^{(n)} | z_1^{(n)} \sim p(z|s), n \leq BE\}$
2: **for** $t = 2$ to $T$ **do**
3:     **for** $b = 1$ to $B$ **do**
4:         Expand the beam $\mathcal{C}_t \leftarrow \{(z_{<t})^{(b)} \circ z_e | z_e \sim p(z_t | z_{<t}^{(b)}, s), e \leq E\}$
5:     **end for**
6:     Decode Trajectories $\mathcal{T}_t \leftarrow \{\tau | \tau = h(z_{<t+1}, s), z_{<t+1} \in \mathcal{C}_t\}$
7:     Get top $B$ trajectories according to return
8:     Set corresponding latent sequences to be $Z_t$
9: **end for**
10: **return** best trajectory in $\mathcal{T}_T$

---

## A    SCALE UP THE PLANNING HORIZON

TAP not only makes it easier to scale up the state and action dimensionality, but also helps the agent to plan over a longer horizon with much less computation. As shown in Figure 4(c), the decision latency increases with the number of planning steps, tested on the hopper-medium-replay task. TAP is able to plan 33 steps into the future within 0.25 seconds. Table 3 shows how the performance on locomotion control tasks varies with the planning horizon. We can see that even just expanding a single latent variable (corresponding to 3 steps in the original space), TAP can actually achieve decent performance in most tasks. The longer horizon planning is helpful for hopper, especially in medium-replay settings. In hopper tasks, the agent can easily fall over compared to other tasks. Once the agent falls over, the episode will be immediately terminated, so similar actions might lead to drastic longer-term differences. This might mean learning the dynamics function and using it to help learn the value function is easier than directly learning the value function. On the other hand, the medium-replay dataset has a higher diversity which may make longer planning useful. Both medium and medium expert datasets are generated by one or two fixed policies. This means the possible actions for the next step are quite limited, so searching may not give very different results given the estimated return. The medium-replay dataset, on the other hand, contains the data in the replay buffer. Given the policy parameters are changing during learning, the behaviour policy is then a mixture of a lot of different policies, making TAP has to consider more options when acting.

## B    THE ROLE OF OOD PENALTY

Besides sampling from the prior distribution given by the transformer, we also used the estimated prior probability to prevent out-of-distribution (OOD) trajectories to be selected during the planning. In Table 4 we showed how the threshold of OOD penalty will affect the performance of TAP. We can

Table 3: Locomotion performance with different planning horizon

| Dataset | Environment | Horizon=3 | Horizon=9 | Horizon=15 | Horizon=21 |
|---|---|---|---|---|---|
| Medium-Expert | HalfCheetah | 91.74 | 91.17 | 91.77 | 90.6 |
| Medium-Expert | Hopper | 88.3 | 104.01 | 105.53 | 96.4 |
| Medium-Expert | Walker2d | 99.4 | 107.25 | 107.44 | 108.82 |
| Medium-Expert | Ant | 126.07 | 130.96 | 128.82 | 134.47 |
| Medium | HalfCheetah | 44.01 | 44.98 | 45.04 | 44.7 |
| Medium | Hopper | 49.7 | 64.44 | 63.44 | 66.7 |
| Medium | Walker2d | 65.04 | 63.21 | 64.87 | 52.03 |
| Medium | Ant | 87.8 | 84.36 | 92.0 | 89.87 |
| Medium-Replay | HalfCheetah | 40.59 | 41.51 | 40.78 | 41.36 |
| Medium-Replay | Hopper | 29.39 | 90.17 | 87.3 | 96.35 |
| Medium-Replay | Walker2d | 57.58 | 58.41 | 66.85 | 69.25 |
| Medium-Replay | Ant | 83.06 | 98.67 | 96.71 | 97.7 |
| **Mean** | | 71.9 | 81.6 | 82.5 | 82.4 |

see when a very low threshold is chosen $\beta = 10^{-5}$, the performance of the agent drops significantly because the trajectory of such low prior probability will be not sampled in the first place and the OOD penalty is not working. Some of the OOD trajectories with high prediction error and high value will be selected in this case. A larger $\beta$ in the range of $[0.002, 0.1]$ gives consistent similar results. However, keeping improving $\beta$ to 0.5 damages the performance again as most of the trajectories do not have such a high prior probability and the penalty will force the agent to choose the most probable trajectory. This $\beta = 0.5$ case can also be treated as an imitation learning baseline, showing searching for the higher return trajectory rather than cloning the behaviour policy is helpful.

Table 4: Locomotion performance with different probability threshold $\beta$

| Dataset | Environment | $\beta = 10^{-5}$ | $\beta = 0.002$ | $\beta = 0.01$ | $\beta = 0.05$ | $\beta = 0.1$ | $\beta = 0.5$ |
|---|---|---|---|---|---|---|---|
| Medium-Expert | HalfCheetah | 64.3 | 91.0 | 90.3 | 91.8 | 92.2 | 87.7 |
| Medium-Expert | Hopper | 66.3 | 95.1 | 94.5 | 105.5 | 107.0 | 76.1 |
| Medium-Expert | Walker2d | 86.6 | 110.1 | 109.2 | 107.4 | 106.5 | 104.0 |
| Medium-Expert | Ant | 104.7 | 132.1 | 133.7 | 128.8 | 127.4 | 116.2 |
| Medium | HalfCheetah | 42.6 | 44.8 | 44.8 | 45.0 | 45.1 | 42.9 |
| Medium | Hopper | 42.9 | 64.6 | 67.0 | 63.4 | 63.8 | 47.5 |
| Medium | Walker2d | 63.4 | 49.2 | 53.8 | 64.9 | 64.4 | 58.7 |
| Medium | Ant | 85.1 | 88.5 | 89.8 | 92.0 | 89.7 | 85.1 |
| Medium-Replay | HalfCheetah | 36.2 | 42.2 | 40.5 | 40.8 | 40.6 | 41.0 |
| Medium-Replay | Hopper | 19.9 | 94.4 | 93.2 | 87.3 | 97.0 | 3.0 |
| Medium-Replay | Walker2d | 27.1 | 57.3 | 59.6 | 66.8 | 65.6 | 50.3 |
| Medium-Replay | Ant | 52.5 | 96.8 | 95.7 | 96.7 | 93.7 | 76.8 |
| **Mean** | | 57.6 | 80.5 | 81.0 | 82.5 | 82.8 | 65.8 |

## C  BASELINES

As for the baselines, we gather the strong baselines for each of the three sets of tasks. We include model-based methods such as MOReL (Kidambi et al., 2020) and (Optimized) MOPO (Yu et al., 2020; Lu et al., 2022); actor-critic methods CQL (Kumar et al., 2020) and IQL (Kostrikov et al., 2022); and sequence modelling methods DT and TT. We use scores reported by the papers by default with a few exceptions.

- All the methods besides TT did not report performance on Ant locomotion control so we run them using their official codebase.
- For AntMaze-Ultra, we run IQL with their official codebase and run $TT_{(+Q)}$ with the code given by the author (Janner et al., 2021).

- The Ant locomotion results of TT come from their official GitHub repository [2].
- Since CQL is tested in the v0 version of locomotion control tasks, we use the v2 results reported by IQL (Kostrikov et al., 2022).
- All the behaviour cloning (BC) results are reported by D4RL (Fu et al., 2020).
- Chen et al. (2021, DT) did not report the results for AntMaze and Adroit so we use DT AntMaze results reported by IQL (Kostrikov et al., 2022).
- MOPO results for Adroit come from (Lu et al., 2022).
- TT results for adroit come from (Wang et al., 2022). We also tested two training seeds, each 10 evaluation episodes, for pen-human and pen-cloned and get $12.9_{\pm 9.2}$ and $17.3_{\pm 9.5}$ respectively.

## D    OTHER DESIGN POTENTIALS FOR THE BOTTLENECK

The design of the prior model is highly conditioned on the design of the bottleneck and the decoder, and will have an impact when used for planning. Following Decision Transformer (Chen et al., 2021) and Trajectory Transformer (Janner et al., 2021), we used a GPT-2 style transformer with causal masking for our encoder and decoder. In this case, the information in the future token will not follow back to recent ones. Such a design is conventional for sequence modelling but not necessarily optimal. For example, one could reverse the order of the masking for the decoder and make the planning goal-oriented, whereas the prior should also be trained in a reversed order.

To fully decouple the planning and temporal structure, we also tried using attention rather than pooling and tiling to construct the bottleneck. Such a design shows a similar performance when doing vanilla sampling. Such a design also prevents us from applying beam search and is thus less efficient and we did not put that to the main text. However, in Table 5 we show TAP with attention bottleneck works as well as fixed length step length indicates the temporal structure of the MDP may not be a necessary inductive bias for planning. This can motivate future works that design more flexible and efficient ways of planning based on the latent-action model.

Table 5: Attention-based bottleneck.

| Dataset | Environment | beam search | $L = 3$ sample | attention sample |
|---------|-------------|-------------|----------------|------------------|
| Medium-Expert | HalfCheetah | 91.8 | 89.9 | 91.3 |
| Medium-Expert | Hopper | 105.5 | 98.5 | 108.2 |
| Medium-Expert | Walker2d | 107.4 | 107.7 | 104.1 |
| Medium-Expert | Ant | 128.8 | 124.7 | 129.3 |
| Medium | HalfCheetah | 45.0 | 44.3 | 45.2 |
| Medium | Hopper | 63.4 | 64.3 | 65.9 |
| Medium | Walker2d | 64.9 | 55.5 | 58.0 |
| Medium | Ant | 92.0 | 88.8 | 90.2 |
| Medium-Replay | HalfCheetah | 40.8 | 39.8 | 39.8 |
| Medium-Replay | Hopper | 87.3 | 79.0 | 79.7 |
| Medium-Replay | Walker2d | 66.8 | 66.0 | 68.8 |
| Medium-Replay | Ant | 96.7 | 81.4 | 89.9 |
| **Mean** | | 82.5 | 78.3 | 80.9 |

## E    ANTMAZE DETAILS

AntMaze is a sparse-reward continuous-control task in which the agent has to control a robotic ant to navigate to the target position. We include the goal position in the observation space so that the generated trajectories are goal-aware.[3] In order to extensively test different methods, we also

---

[2]https://github.com/JannerM/trajectory-transformer

[3]The goal position is specified by the `infos/goal` field in the d4rl dataset by default.

add a customized larger antmaze that we called Antmaze-Ultra, which has doubled the size of the previously largest map (Antmaze-Large).

Antmaze has the same dimensionality as locomotion ant and so will not provide too much direct evidence about the action dimension scalability, so it's more about an orthogonal evaluation on sparse reward problems. AntMaze is challenging because there are a lot of sub-optimal trajectories in the dataset that is navigating to different goal positions rather than the target position in the test time. Reaching these goals will not give a reward to the agent. The reward will only be given when the agent reaches the true target, which is also the goal in the test time. On AntMaze, TAP shows better performance on a more challenging large dataset, but is slightly inferior to $\text{TT}_{(+Q)}$ on the medium datasets. We did not use the IQL critic as our value estimator; a better value estimation approach than our current Monte-Carlo estimation would be expected to bring orthogonal improvements. Effective performance without a separate $Q$-function is partially due to the inclusion of the goal position into the observation: TAP can learn to generalize across goal positions instead of completely relying on the value estimation.

TT has shown using a separate IQL value function can help sequence generative models to solve AntMaze. Such an approach, however, further increases the computational cost for sampling trajectories. Here we instead propose an alternative efficient approach that let TAP jointly leverage trajectories with different objectives and also the reward signals. This is as simple as putting the goal positions to the observation of the agent. We hypothesized that goal positions can be helpful for TAP as it models the whole distribution of the trajectories. When acting in the environment, having the Trajectories conditioned on the goal can narrow down the sampled trajectories to focus on the correct direction, therefore simplifying the planning. Note that for actor-critic methods, the positions of the test time target position are encoded by the critic network and including the goal position will not provide extra information to the agent.

In order to pressure test IQL, $\text{TT}_{+Q}$ and TAP in larger AntMaze environments, we introduced the AntMaze-Ultra task, which has $10 \times 14$ blocks of effective size. This is 4 and 2 times as large as AntMaze-Medium ($6 \times 6$) and AntMaze-Large ($7 \times 7$) tasks, respectively. A visualisation of these three tasks can be found in Figure 5. In addition to the increased size, the complexity of the walls also results in multiple possible routes to navigate from the bottom left corner to the top right. Similar to the antmaze-medium and antmaze-large tasks, we let the agent run for 1K episodes, each with 1K steps to collect the trajectories. However, we find that such an amount of data is far away from being enough to learn a proper value function, as IQL gives a very poor performance on this task. Using the Monte-Carlo estimation, TAP even suffers more from this issue. We find that planning with beam search, in this case, is even worse than just sampling a random trajectory: with random sampling, TAP gives 22.00 +/- 4.14 on AntMaze-Ultra-Play and 26.00 +/- 4.39 on AntMaze-Ultra-Diverse. However, beam search will decrease its performance to 21.00 +/- 4.07 on Ultra-Play and 22.00 +/- 4.14 on Ultra-Diverse.

Table 6: Antmaze results. The Antmaze-Ultra is our customised larger environment.

| Dataset | Environment | BC | CQL | IQL | DT | $\text{TT}_{(+Q)}$ | $\text{TAP}_{(+G)}$ |
|---|---|---|---|---|---|---|---|
| Play | Antmaze-Medium | 0.0 | 61.2 | 71.2 | 0.0 | **93.3** $\pm 6.4$ | 78.0 $\pm 4.14$ |
| Diverse | Antmaze-Medium | 0.0 | 53.7 | 70.0 | 0.0 | **100.0** $\pm 0.0$ | 85.0 $\pm 3.57$ |
| Play | Antmaze-Large | 0.0 | 15.8 | 39.6 | 0.0 | 66.7 $\pm 12.2$ | **74.0** $\pm 4.39$ |
| Diverse | Antmaze-Large | 0.0 | 14.9 | 47.5 | 0.0 | 60.0 $\pm 12.7$ | **82.0** $\pm 5.00$ |
| Play | Antmaze-Ultra | – | – | 8.3 | – | 20.0 $\pm 10.0$ | **22.0** $\pm 4.1$ |
| Diverse | Antmaze-Ultra | – | – | 15.6 | – | **33.3** $\pm 12.2$ | 26.0 $\pm 4.4$ |
| | **Mean** | - | - | 42.0 | - | **62.2** | **61.2** |

## F  INTERPRET LATENT CODES

A key design choice we made is to let the decoder, not only the prior of latents, be conditioned on the current state. Such a design leads to a compact latent action space that enables efficient planning. Without the state as input, the decoder must reconstruct the whole trajectory relying solely on the latent codes, so the latent codes contain information about the whole trajectory segments

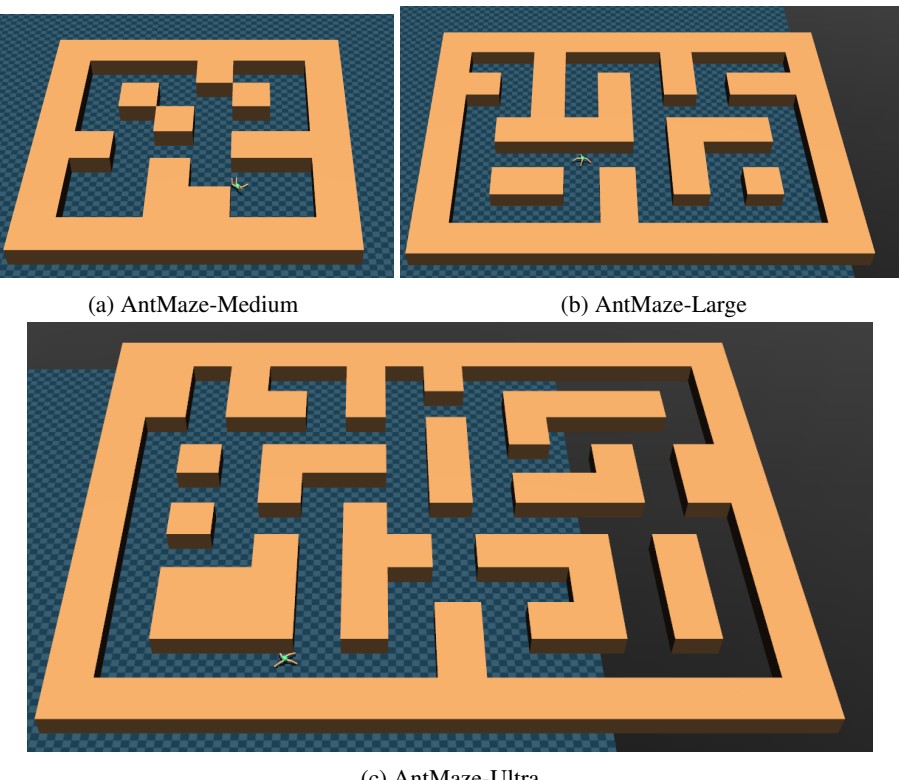

(a) AntMaze-Medium

(b) AntMaze-Large

(c) AntMaze-Ultra

Figure 5: Visualization of different antmaze environments.

(states, actions, rewards, values). In other words, latent codes will be a discrete representation of the trajectory, where a latent codes sequence can be decoded into a unique trajectory. On the other hand, conditioned on the current state, the latent codes only need to carry information about how the future trajectories branch from the existing trajectory. Every $L$ step, a branch of the trajectory will fork into $K$ possible futures. Each of the latent codes corresponds to a branch of this tree of possible future trajectories, where the root node is the current state. The forking can be caused by the uncertainty of the dynamics model or by the stochasticity of the behaviour policy. The number of latent codes needed to reconstruct a trajectory given a starting state will reduce as long as the model knows more about the dynamics of the environment and the behaviour policy. Assume a trajectory distribution $p(\tau|\boldsymbol{s}_1; \mu)$ with behaviour policy $\mu$ has deterministic Markovian transition dynamics, and the model has recovered the true dynamics. The probability of a valid trajectory that fit the dynamics can be expressed as $p(\tau|\boldsymbol{s}_1; \mu) = \prod_{i=1}^{T} \mu(\boldsymbol{a}_i|\boldsymbol{s}_i)$. So the discrete latent variables only need to approximate the distribution of the behaviour policy, where latent codes correspond to grouped components of the behaviour policy. Note that the policy is not only grouped in a single step but also across multiple steps. Intuitively, the latent codes can then be interpreted as state-conditioned options (Sutton et al., 1999), learning by decomposing the behaviour policy into a weighted sum of sub-policies. The number of grouped policy components of the behaviour policy can be much less than the dimensionalities needed to represent the whole trajectory. As we will show in Section 4.4, the $L = 3$ or $L = 4$ gives an optimal performance of TAP on D4RL tasks, which means 3 steps of the transitions with up to hundreds of dimensions can be expressed with a single discrete latent variable, with $K = 512$ bins. In contrast, the discrete representation of the trajectory used in TT will allocate $LD$ discrete codes to describe the same number of transitions. This means TAP has a smaller and more compact latent space for downstream planning.

## G  HYPERPARAMETERS

In Table 7 we showed the hyperparameters for TAP, both for training and planning.

Table 7: List of Hyper-parameters

| Environment | Hyper-parameters | Value |
|---|---|---|
| All | learning rate | $2e^{-4}$ |
| All | batch size | 512 |
| All | dropout probability | 0.1 |
| All | number of attention heads | 4 |
| All | number of steps for a latent code | 3 |
| All | beam expansion factor | 4 |
| All | Embedding size for a latent code | 512 |
| All | $\beta$ | 0.05 |
| Locomotion Control | training sequence length | 24 |
| Locomotion Control | discount | 0.99 |
| Locomotion Control | number of layers | 4 |
| Locomotion Control | feature vector size | 512 |
| Locomotion Control | $K$ | 512 |
| Locomotion Control | beam width | 64 |
| Locomotion Control | planning horizon | 15 |
| AntMaze | training sequence length | 15 |
| AntMaze | discount | 0.998 |
| AntMaze | number of layers | 4 |
| AntMaze | feature vector size | 512 |
| AntMaze | $K$ | 8192 |
| AntMaze | beam width | 2 |
| AntMaze | planning horizon | 15 |
| Adroit | training sequence length | 24 |
| Adroit | discount | 0.99 |
| Adroit | number of layers | 3 |
| Adroit | feature vector size | 256 |
| Adroit | $K$ | 512 |
| Adroit | beam width | 256 |
| Adroit | planning horizon | 24 |

Table 8: The ablation of number of steps for a latent code $L$.

| Dataset | Environment | $L = 1$ | $L = 2$ | $L = 3$ | $L = 4$ |
|---|---|---|---|---|---|
| Medium-Expert | HalfCheetah | 37.4 | 68.1 | 91.8 | 92.5 |
| Medium-Expert | Hopper | 44.7 | 55.6 | 105.5 | 102.2 |
| Medium-Expert | Walker2d | 94.9 | 108.9 | 107.4 | 108.5 |
| Medium-Expert | Ant | 122.7 | 118.0 | 128.8 | 132.2 |
| Medium | HalfCheetah | 43.5 | 42.6 | 45.0 | 45.4 |
| Medium | Hopper | 55.1 | 70.5 | 63.4 | 70.7 |
| Medium | Walker2d | 35.2 | 70.6 | 64.9 | 69.3 |
| Medium | Ant | 92.3 | 98.2 | 92.0 | 88.9 |
| Medium-Replay | HalfCheetah | 30.3 | 30.1 | 40.8 | 40.0 |
| Medium-Replay | Hopper | 61.9 | 69.9 | 87.3 | 85.6 |
| Medium-Replay | Walker2d | 29.4 | 60.5 | 66.8 | 54.8 |
| Medium-Replay | Ant | 80.1 | 90.0 | 96.7 | 95.1 |
| **Mean** | | 60.6 | 73.6 | 82.5 | 82.1 |

## H   PREDICTION ERROR, SEARCH VALUE AND PERFORMANCE

To investigate the reason that temporal abstraction and state-conditional encoder will be helpful for TAP, we provide more metrics about TAP search in Figure 6. Scores show the performance of the agent. Search values are optimal value(return) found by TAP in the initial states. Errors are mean squared errors of state prediction given the action sequences output by the TAP. We can see: 1) For state-conditional decoder, there is a negative correlation between prediction errors and scores.

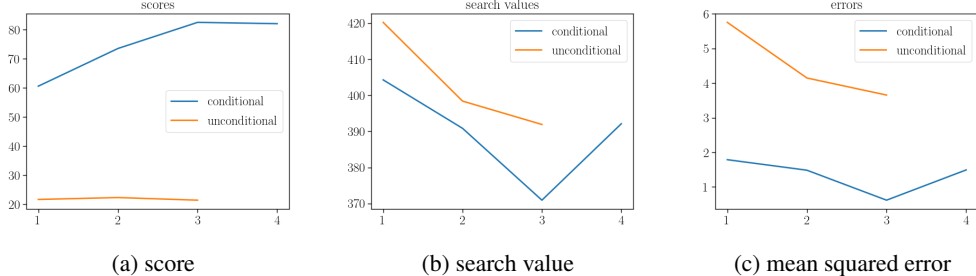

| (a) score | (b) search value | (c) mean squared error |

Figure 6: Scores, search value and error for TAP agent with different latent step $L$ and with/without state-conditional encoder.

This not surprising because better prediction accuracy can naturally lead to better performance. 2) For both state-conditional and unconditional decoder, search values are positively correlated to the errors. This indicates extra value improvement might be caused by over-optimistic prediction. 3) Both search value and errors are minimal at $L = 3$. Our interpretation about this is: Smaller $L$ can lead to overfitting in the training data, therefore worse generalization and finer-grained search, which will amplify generalization error. On the other hand, larger $L$ can cause underfitting therefore higher error. 4) Naively increasing the number of latent variables is not helpful for the unconditional decoder.

Table 9: Comparing beam search against direct sampling from the prior policy.

| Dataset | Environment | beam search | sample prior | sample uniform |
|---|---|---|---|---|
| Medium-Expert | HalfCheetah | 91.8 | 89.9 | 41.8 |
| Medium-Expert | Hopper | 105.5 | 98.5 | 62.3 |
| Medium-Expert | Walker2d | 107.4 | 107.7 | 86.7 |
| Medium-Expert | Ant | 128.8 | 124.7 | 105.4 |
| Medium | HalfCheetah | 45.0 | 44.3 | 39.5 |
| Medium | Hopper | 63.4 | 64.3 | 39.6 |
| Medium | Walker2d | 64.9 | 55.5 | 70.2 |
| Medium | Ant | 92.0 | 88.8 | 89.8 |
| Medium-Replay | HalfCheetah | 40.8 | 39.8 | 10.2 |
| Medium-Replay | Hopper | 87.3 | 79.0 | 14.7 |
| Medium-Replay | Walker2d | 66.8 | 66.0 | 7.8 |
| Medium-Replay | Ant | 96.7 | 81.4 | 47.0 |
| **Mean** | | 82.5 | 78.3 | 51.3 |

## I  SAMPLED DISTRIBUTIONS

We sampled 2048 trajectories either uniformly or according to the prior in the first step of the 10 evaluation episodes and plot their metrics in Figure 7. The x-axis of the plots is predicted returns by the agents. The y-axis shows the state's prediction mean-squared error (MSE), where the ground truth is given by the simulator following the action sequences of the predicted trajectories. Sampling from the prior does not damage the diversity of the sampled trajectories in terms of the variance of returns. On the other hand, the expected MSE is reduced in both hopper and ant cases. This means that by sampling from the prior, we can get a higher quality of trajectory samples. We can see that TAP successfully approximates the continuous distribution of returns and the generated trajectories are high in diversity in terms of the returns. This is especially true for hopper-medium-replay because the behaviour policy itself is of high diversity, and we can see multiple modes in the distribution. It is interesting to see that the trajectories with higher predicted returns do not necessarily have higher prediction errors. This is a nice feature because we can search for high-return trajectories without worrying too much about these trajectories being unrealistic.

Table 10: State unconditional decoder with different length steps.

| Dataset | Environment | $L = 1$ | $L = 2$ | $L = 3$ |
|---------|-------------|---------|---------|---------|
| Medium-Expert | HalfCheetah | 10.0 | 11.0 | 9.5 |
| Medium-Expert | Hopper | 27.8 | 29.6 | 42.6 |
| Medium-Expert | Walker2d | 50.0 | 46.9 | 15.7 |
| Medium-Expert | Ant | 32.5 | 30.9 | 26.1 |
| Medium | HalfCheetah | 18.2 | 16.5 | 13.6 |
| Medium | Hopper | 37.0 | 39.2 | 47.0 |
| Medium | Walker2d | 13.6 | 17.3 | 24.8 |
| Medium | Ant | 18.5 | 21.5 | 19.7 |
| Medium-Replay | HalfCheetah | 4.8 | 5.7 | 5.0 |
| Medium-Replay | Hopper | 19.6 | 16.6 | 22.6 |
| Medium-Replay | Walker2d | 8.2 | 8.9 | 9.6 |
| Medium-Replay | Ant | 20.0 | 24.2 | 21.1 |
| **Mean** | | 21.7 | 22.4 | 21.4 |

(a) hopper uniform     (b) hopper prior     (c) ant uniform     (d) ant prior

Figure 7: Distribution of trajectory returns and state prediction mean-squared errors. The dataset for the hopper is medium-replay and that for ant is medium.

In the ablation of sampling from prior versus sampling from a uniform distribution, we saw sampling from the prior policy gives much better performance. Comparing the distribution of returns and errors of trajectories sampled from the prior and uniform distribution, we observe the sampling from prior policy yields trajectories with lower reconstruction error but with similar returns. This indicates sampling from prior will improve the quality of trajectories being generated from TAP, which explains better performance.

## J  TRAINING COST

The training time of TAP and TT are similar for lower dimensionality, TT needs 6-12 hours and TAP needs 6 hours for gym locomotion control tasks, on a single GPU. It's worth noting that the training cost of TT will also grow quickly along the dimensionality. For adroit, the same training for TT takes 31 hours, but the training cost of TAP is the same given the same architecture. In fact, for adroit experiments with TAP, we used a smaller network (see Table 7 for hyper-parameters) which needs fewer training epochs and the whole training can be finished within 1 hour.

## K  VISUALIZE PLANS

To give an intuitive understanding of the latent action space, predicted trajectories and planning of TAP. In we show a visualization of the generated plans by both direct sampling and beam search with 256 samples (beam width=64 and expansion factor=4 for beam search). Each frame shows 256 latent codes and the corresponding state in a particular step in the plan, where all the plans start from an initial state of the hopper task. The trajectories are sorted according to the objective score,

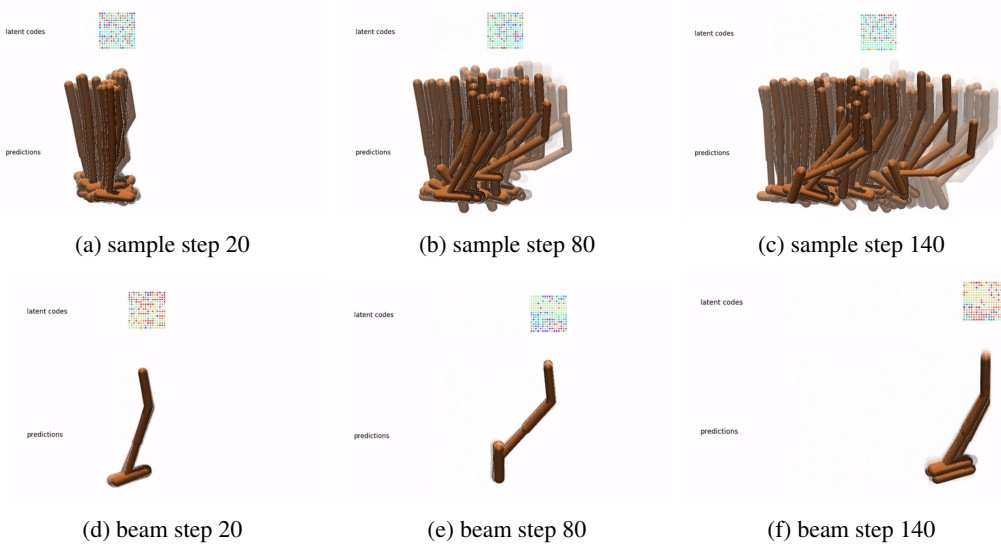

Figure 8: Visualization of latent codes, decoded trajectories and planning.

so that the most front-end trajectory will be the plan to be executed. Direct sampling generates more diverse trajectories while most of them are of low prior probability. There are some of the predicted trajectories (opaque) that move quickly but do not follow the environment dynamics. The plan that is chosen follows the true dynamics but is suboptimal as it's falling down. On the other hand, beam search generates trajectories that are both with high returns and high probability. The model predicts 144 steps of the future and is trained on hopper-medium-replay.

