# OpenReview forum: "Efficient Planning in a Compact Latent Action Space"
_ICLR.cc/2023/Conference — ICLR 2023 poster_

### Official Review · Reviewer_Ykqw · 2022-10-23

**Confidence:** 4
**Correctness:** 3
**Technical Novelty And Significance:** 3
**Empirical Novelty And Significance:** 2
**Recommendation:** 5

**Clarity, Quality, Novelty And Reproducibility:**

Clarity: very clear. great writing!

Quality: solid execution of an intuitive idea, although the evaluation task complexity can be improved.

Originality: somewhat questionable. See the weakness comments on missing references.

**Strength And Weaknesses:**

Overall I quite like the overall premise of the paper: if learning and planning with an accurate and comprehensive model for a control problem is hopelessly complex and expensive, why not focus on the part of the problem that is relevant to the task? I think learning a latent action space and building a constrained dynamics model on top is a nice way to approach this problem. And the proposed method is conceptually simple and easy to dissect. I also like how comprehensive the ablation study is.

However,  I believe there are important issues with this paper that need to be addressed. The issues are (1) missing references that are highly relevant and (2) conceptual and empirical differences between learning models and learning behaviors.

1. Missing references
There are a number of important references that are missing. First, there is an entire set of work in hierarchical imitation learning [1][2] that tries to learn temporally-abstracted behavior primitives and use them for planning / more efficient imitation. For example, Mandlekar et al., [1] uses VAE to learn a latent space of short-horizon goal-conditional policies and use that for value-based planning. Similarly, [2] learns a latent space of short behaviors and uses them for one-shot imitation. From a practical point of view, these behavior primitives and how they are used are no different than the latent action proposed in this work. In the context of more RL-ish works, Allshire and Martin-Martin et al. [3] is highly relevant. Similar to the premise and motivation of this paper, they learn a latent action space based on offline experience and use it for more efficient RL. There is also [4], which has similar line of argument. Although they do not study temporal abstraction, it is worth discussing and contextualizing the proposed method with this work. Finally, in a slightly far away domain, a number of works [5][6] proposed to learn temporally-abstract action space for better teleoperation. I hope the authors can select a subset of these works and their predecessors / followup works and provide a more thorough account of the idea of learning task-relevant action space, instead of just focusing on offline & model-based RL. In addition, some of these works may be directly comparable and may serve as strong empirical baselines, e.g., [1].

[1] IRIS: Implicit Reinforcement without Interaction at Scale for Learning Control from Offline Robot Manipulation Data, Mandlekar et al., 2019
[2] Learning latent plans from play data, Lynch et al., 2019
[3] LASER: Learning a Latent Action Space for Efficient Reinforcement Learning, Allshire and Martin-Martin et al, 2021
[4] PLAS: Latent Action Space for Offline Reinforcement Learning, Zhou et al., 2020
[5] Learning latent actions to control assistive robots, Losey et al., 2022
[6] Learning visually guided latent actions for assistive teleoperation, Karamcheti et al., 2021

2. Learning models vs. learning behaviors.
As mentioned in my synopsis of this work, I believe the proposed method is more or less limited to learning from offline datasets that is generated by an external behavior policy, since it assumes there exists such a task-relevant state-action space to learn the latent action. So how does one differentiate the proposed “model-based offline RL” methods from conditional imitation learning? Based on mainstream understanding, the main advantage of model-based approach is that the model can be shared among many different tasks, as the per-step dynamics are agnostic to a task goal. And one may use the same model to synthesize behaviors that can achieve new goals in a zero-shot manner. It’d be interesting to see how much of the “model-based” quality the proposed method still retains. Concretely, I’d like to see experiments that focus on generalization to new goals and behaviors.

My final critique that is not as important as (1) and (2) is the lack of more complex evaluation task. Since one of the main argument that the paper makes is computational efficiency, I'd really wish to see some real-robot experiment to substantiate this claim. Along the same line, it'd be great if the authors could at least evaluate on simulated benchmark such as robomimic (Mandlekar et al., 2021) for which the tasks are directly modeled after setup that can be replicated in the physical world.

**Summary Of The Paper:**

The paper describes a model-based reinforcement learning method designed to learn from offline datasets (generated by an external behavior policy). The method attempts to address two key challenges in model-based RL. First, for common continuous control problems such as locomotion and manipulation, the action space is high-dimensional and fine-grained, resulting in an expensive search problem. Second, learned environment models are often riddled with “holes” (regions with poor dynamics & cost estimates) due to uneven data coverage. Uniform trajectory sampling strategies may easily fall into these holes and yield poor performance. The proposed method addresses the first problem by learning a latent action space, where each discrete latent action maps to a short snippet of trajectory. The computational cost of searching in such action space can be adjusted using the trajectory snippet length. The latent action space is discrete and finite and is learned end-to-end with a VQ-VAE (VAE with discrete priors) trajectory reconstruction objective. Such data-informed action space also partially addresses the second problem (holes), as the search is naturally confined to trajectory patterns that exist in the training set. To further tighten the search space, the method proposes to learn the likelihood of the next latent action conditioning on the first state and the latent actions in the plan so far. The method also employs beam search to further improve the planning efficiency. The method is evaluated on a simulated offline RL benchmark (D4RL), which includes continuous control problems such as half-cheetah, and tasks with higher dimensional action space (dexterous manipulation). The result shows that the proposed method achieves comparable performances relative to a number of model-based and model-free RL baselines for tasks with low-dimension actions, and compares favorably in tasks with higher-dimensional action space.

**Summary Of The Review:**

Again, I like the overall premise and the high-level idea of the work. And I think the method is conceptually simple and would be easy to work with. I'd like to see an in-depth discussion of missing references listed above and also addressing the gap between learning environment models and learning behaviors. More complex evaluation tasks can also substantially improve the overall quality of the paper.

---

> ### Author Response · Authors · 2022-11-14
> **Individual Response (1)**
>
> Thanks for your insightful comments and constructive feedback! We’ve updated the paper and then try to address your main concerns as follows. We hope you will agree this makes the paper stronger and addresses your concerns, and that you will consider raising your score in response.
>
> ## Missing references
> Thanks for your reminder of related work. We’ve updated the related work which now includes all the literature you’ve mentioned. Please let us know if the current related work works well for you.
>
> We summarised the key novelty that differentiates our work from all previous methods in the general response and the detailed differences with each individual work in the following:
>
>
> **IRIS: Implicit Reinforcement without Interaction at Scale for Learning Control from Offline Robot Manipulation Data**
>
> IRIS tries to address an imitation learning setting where there are suboptimal trajectories in the demonstration. To achieve that, IRIS trains a VAE to model the distribution of future states $p(s’|s)$ and use that to generate intermediate goals which are then filtered by a value function. Once a goal is selected, the agent will be driven by a goal-reaching policy.
>
> This work is relevant because, on a high level, TAP can be reinterpreted into stitching segments of existing trajectories to produce a better one, which is similar to the concept of hierarchical imitation in IRIS. From that perspective, TAP planning can be loosely linked to the IRIS goal sampling with value filtering and action reconstruction can be mapped to goal-conditioned policy. Whereas we also want to push back that such a high-level metaphor can be applied to almost all the offline model-based methods. With out-of-distribution action constraints, a dynamics model plus a policy can be regarded as a generative model of value-filtered training data, where any planning can be goal sampling and following the plan thus corresponding to goal-conditioned policy.
>
>
> **Learning latent plans from play data, Lynch et al., 2019**
>
> Play-LMP addresses a problem in goal-oriented imitation learning where there the path between the current state and the goal state can be multimodal, which is difficult to parameterise in the continuous case. Therefore, they train a VAE to disentangle different paths between two states and have the goal-reaching policy to be conditioned on the latent representation of the path.
>
> This work is remotely relevant because both TAP can Play-LMP uses (variations) of VAE to parameterise the distribution of the trajectories. If we also treat the decoder of TAP as a policy it also takes the current state and latent vectors as inputs and outputs raw actions, similar to the goal and latent-conditioned policy in Play-LMP.
>
> **General Differences between TAP and Goal-oriented imitation (IRIS and Play-LMP)**
>
> We believe these two works are only related to TAP on a very high level given the implementation is drastically different. On a conceptual level:
>
> First of all, TAP operates in an RL setting where we care about the sum of rewards along the trajectory, not just reaching a particular goal state.
>
> Secondly, TAP learned a structured discrete latent space with a VQ-VAE, which allows efficient multi-step planning in the latent space. But the goal selection for IRIS is just brutal sampling and the term “plan” in Play-LMP seems to be an alias for trajectory or action sequences, not a process of optimization.
>
> **LASER: Learning a Latent Action Space for Efficient Reinforcement Learning, Allshire and Martin-Martin et al, 2021**
>
> **PLAS: Latent Action Space for Offline Reinforcement Learning, Zhou et al., 2020**
>
> These works are relevant because they also learned a latent action space and the policy optimization happens in the latent action space.
> The differences are these works are all model-free which doesn’t have a predictive model, and there is no decoupling between decision steps and the environment's temporal structure, namely, no temporal abstraction.
> The motivation for latent action is also different because, for TAP, the purpose of learning a discrete latent action space is to enable efficient planning. TAP jointly learns the structured compact latent space and the model by training a trajectory-level VQ-VAE, where actions correspond to sub-policies. While model-free methods usually construct a continuous representation of actions without temporal abstraction.
>
>
> **Learning latent actions to control assistive robots, Losey et al., 2022**
>
> **Learning visually guided latent actions for assistive teleoperation, Karamcheti et al., 2021**
>
> The methodology of these works is similar to the model-free RL ones besides they aim to provide a better interface for human manipulation. Another similarity to TAP is they also introduced temporal abstraction.
> The differences are obvious because high-level decision-making is done by a human, not an automated agent.

---

> > ### Author Response · Authors · 2022-11-14
> > **Individual Response (2)**
> >
> > ## Learning models vs. learning behaviors
> > > As mentioned in my synopsis of this work, I believe the proposed method is more or less limited to learning from offline datasets that is generated by an external behavior policy, since it assumes there exists such a task-relevant state-action space to learn the latent action… (for conventional model-based RL) as the per-step dynamics are agnostic to a task goal. And one may use the same model to synthesize behaviors that can achieve new goals in a zero-shot manner
> >
> > To avoid misinterpretation of the point here, here is our understanding of your position (please let us know if our understanding is wrong): you say that because latent actions only capture in-distribution actions in the dataset, it’s likely that TAP cannot generalize to new goals which require novel action sequences.
> >
> > On one hand, we agree TAP cannot query a novel action that is far out of distribution. However, querying such actions will actually be harmful when the model is learned from data.
> > On the other hand, going beyond the expert trajectory or generalising to novel tasks can be done by recombing in-distribution policies. TAP can capture a mixture of diversed policies just like the case in medium-replay dataset in d4rl gym-locomotion as visualized in updated Appendix K, Figure 8. In principle, TAP can zero-shot generalize to novel tasks by recombining latent actions, guided by a proper objective function.
> >
> >
> > > So how does one differentiate the proposed “model-based offline RL” methods from conditional imitation learning?
> >
> > I assume conditional imitation learning means return-conditioned behaviour cloning (BC) in this context. We believe one of the important differences is TAP **learned the reward structure** and can stitch existing behaviours to get a policy that goes beyond behaviour policy, whereas behaviour cloning can only try to reproduce expert behaviours.
> >
> > There are 2 pieces **empirical of evidence** that we think support the claim that TAP is not only trying to recover the best trajectory seen in the data:
> > 1. For Adroit-expert datasets, TAP managed to surpass the performance of expert performance (100 means the same as the expert).
> > 2. Ablation of the planning horizon: longer planning steps are helpful for the performance of TAP. If TAP is only doing imitation and no action sequence recombination is needed then a single step of prediction should be sufficient since the (on-policy) return already captures the information about future trajectories.
> >
> > > Based on mainstream understanding, the main advantage of model-based approach is that the model can be shared among many different tasks, ... It’d be interesting to see how much of the “model-based” quality the proposed method still retains. Concretely, I’d like to see experiments that focus on generalization to new goals and behaviors.
> >
> > We agree transfer learning will be an exciting direction for the future work of TAP. Especially if we can train TAP on a large and diverse multi-task dataset, ideally generated by human activities (e.g. motion capture, computer usage). The latent actions can then capture low-level skills that can be reused by multiple tasks and the decoder is a predictive model that takes these latent actions as inputs.
> >
> > Empirically, we believe **antmaze experiments in the Appendix E Table 6** is a proof-of-concept of generalization since the training time goal and initial positions are randomized. In this task, all model-free methods (CQL, IQL, Decision Transformer) struggle but model-based ones (TAP, TT) achieves strong performance.
> >
> > On the other hand, we are not sure if the RL community will agree that zero-shot transfer is the only advantage of model-based methods. In single-task settings, planning-based methods (in particular muzero and its variations) maintained state-of-the-art for board games, Atari and openai procgen environments. As for the benefits of having a model, [1] shows planning at the test time can be helpful for some strategic tasks (e.g. board games, Sokoban) and for other more reactive tasks planning can also be used to construct a much stronger policy improvement and value estimation operator (e.g. Atari games). [2] also shows learning a model itself can be beneficial for representation learning that can further improve the performance of the learned policy.
> >
> > [1] On the role of planning in model-based deep reinforcement learning, Hamrick et al
> >
> > [2] Muesli: Combining Improvements in Policy Optimization, Hessel et al

---

### Official Review · Reviewer_Brj7 · 2022-10-25

**Confidence:** 3
**Correctness:** 4
**Technical Novelty And Significance:** 3
**Empirical Novelty And Significance:** 3
**Recommendation:** 6

**Clarity, Quality, Novelty And Reproducibility:**

* The paper is written clearly and easy to follow.
* The idea of learning and planning in the temporal abstracted latent action space is novel. Applying the overall method to Offline RL setting is reasonable.
* The code is also attached in the supplementary material. So, I guess the experimental results can be reproduced properly.

**Strength And Weaknesses:**

**Strength:**
  *  Using Transformer based VQ-VAE to achieve both *spatial abstraction* and *temporal abstraction* can significantly reduced the search space of the planning based RL method.
  * Compared with Trajectory Transformer, considering each transition tuple $(s_t, a_t, r_t, R_t)$ as a single token for the Transformer is much more reasonable, which could significantly reduce the computational cost.
  * The paper is written clearly and easy to follow.


**Weaknesses:**
  * While using compact action representation could reduce the search space, the model also losses the ability of doing more fine-grained control and can potentially lead to sub-optimal action selection. How to balance the two?
  * How to apply TAP to the online RL setting where we do not have lots of and diverse existing data to train the action encoder?
 * As the authors also mentioned in the limitation part, TAP currently cannot not distinguish between uncertainty stemming from a stochastic behaviour policy and uncertainty from the environment dynamics.
  * It's better to provide the overall training loss function as well.
  * [1] may be a missing related work.
  * Dreamer V2 [2] also use discrete latent representations. It's better to discuss connection between the two.
  * Minors:
    * "After adding the positional embedding, the encoder then reconstructs the trajectory",  the 'encoder'  should be 'decoder'.
    * "As such, the planning process only needs to optimize in a 5 or 8-dimensional discrete space with a learned prior", the 'discrete space' should be 'planning horizon'.



**Reference**

[1] Continuous Control with Action Quantization from Demonstrations

[2] Mastering Atari with Discrete World Models

**Summary Of The Paper:**

To alleviate the computational overhead of the planning-based RL algorithms, this paper proposes an Offline RL method Trajectory Autoencoding Planner (TAP), which learns and plans in the compact latent action space.
The key points are:
  * (1) TAP achieves both *spatial abstraction* and *temporal abstraction* by applying a Transformer based VQ-VAE to encode each transition tuples $(s_t, a_t, r_t, R_t)$ in a trajectory $\tau$ into a latent space and then merging every $L$ adjacent encoded transtions into one latent action in the discrete codebook of VQ-VAE.
  * (2)  TAP uses a Transformer to autoregressively model the prior distribution of the latent action codes.
  * (3) The planning is executed in the discrete latent action space by applying the trained prior latent action sampler and beam search. As multiple environmental actions are aggregated into one latent action, the planning horizon is reduced to $1/L$.


**Summary Of The Review:**

Overall, this paper proposes an interesting idea of learning and planning in the temporal abstracted latent action space, which significantly reduces the computational cost compared to its counterpart Trajectory Transformer. I recommend an accept.

---

> ### Author Response · Authors · 2022-11-14
> **Individual Response**
>
> Thanks for your support of our work! Here’s the response to some of your questions.
>
> > While using compact action representation could reduce the search space, the model also losses the ability of doing more fine-grained control and can potentially lead to sub-optimal action selection. How to balance the two?
>
> That’s a good question. An interesting finding during the ablations is the performance of the agent and search space is not contradictory to each other, because fine-grained control does not always bring better performance. An easy way to increase the control granularity is to introduce more latent code but it doesn’t seem to bring any benefits in D4RL empirically (**see ablation study, Figure 3 and Table 8**). We believe when the size of the dataset is small, having fine-grained control might be harmful since the planner has more choices to exploit the inaccuracy of the model.
>
> > How to apply TAP to the online RL setting where we do not have lots of diverse existing data to train the action encoder?
>
> Solving online RL involves two sub-problems: policy improvement and exploration.
> Our experiments in the offline setting have shown TAP can do proper policy improvement, where the medium-replay datasets are exactly the same as the replay buffer of an online off-policy RL algorithm.
>
> As for exploration, most online RL algorithms rely on noise centred around the current best policy, which can also be added to the raw actions decoding process.
> When having a diverse dataset, the latent action space can even bring extra benefits for exploration. One can consider deliberately choosing latent actions with lower probability, which might lead to even more structured planning.
>
> > Dreamer V2 [2] also use discrete latent representations. It's better to discuss the connection between the two.
>
> We cited Dreamerv2 in the related work “planning in the Latent Space” without explicitly mentioning the name Deamerv2. The key difference is TAP learns and plans a discrete action space and dreamverV2 learned a discrete state space.
>
> > [1] Continuous Control with Action Quantization from Demonstrations
>
> Thanks for pointing that out! I’ve added a new subsection in the related work to discuss model-free RL methods with latent action space.
>
> Please let us know if these address your concerns, or if there are any further concerns preventing you from increasing the score to 8.

---

> > ### Comment · Reviewer_Brj7 · 2022-12-05
> > **Response to authors**
> >
> > Thank the authors for the detailed response. I keep my score for now.
> >
> > I agree with reviewer *rxeb* that conducting one more ablation study to verify the effectiveness of the learned temporal abstraction would be better: modifying TAP by setting L=1, i.e., only using VQ-VAE to discretize the action space and simply using the *action repeat* trick (i.e. executing the same action for multiple steps) to reduce the planning horizon.
> >
> > Besides, I see one more interesting discussion about the action dimensionality of Go and Chess. My point is that we cannot say that the action space of Go is much smaller than the continuous control problems just because the stone of Go is put on a 2-dimensional board. The action space of Go is naturally discrete and the difficulty of the game depends on the number of grids on the board. In Go, it is obvious that the Values of different positions are significantly different (i.e., the curve of the value function changes drastically), but the curves of the action-value functions in continuous control tasks are smooth, continuous, and changing slowly.

---

> > > ### Author Response · Authors · 2022-12-05
> > > **Ablation on temporal abstraction and discussion on action dimensionality**
> > >
> > > Thanks for your response! To discuss the new points you raised:
> > >
> > > > one more ablation study to verify the effectiveness of the learned temporal abstraction would be better: modifying TAP by setting L=1,
> > >
> > > Our ablation in Figure 3 (yellow bars) and Table 8 gives information about how temporal abstraction will affect the performance of TAP. Basically, having L=1 will not only **increase the planning time** but also **reduce the performance**. In Appendix Figure 6 we show this is caused by over-optimistic plans: finer grain latent actions can cause overfitting during training and it's also easier for the planner to exploit the model during planning.
> > >
> > > > i.e., only using VQ-VAE to discretize the action space and simply using the action repeat trick (i.e. executing the same action for multiple steps) to reduce the planning horizon.
> > >
> > > While we think our ablation already shows how temporal abstraction can affect TAP, we believe the alternative you proposed here is also interesting.
> > > There are a few interpretations of "only using VQ-VAE to discretize the action space", we interpret it as using VQ-VAE to discretize the action space, not the state/trajectory space. This one makes sense and it will be quite different from TAP because
> > > 1) using VQ-VAE to discretize the action space clusters the actions while TAP clusters the behaviour policies.
> > > 2) "prior" transformer will also predict states in the continuous space
> > > 3) loss of the ability to decouple the planning steps and environment steps (as you suggested)
> > >
> > > An alternative interpretation is to do the same thing as our ablation did.
> > >
> > > About action repetition, we agree it can help reduce the planning time by $\frac{1}{L}$ while it introduces delayed responses to the new states which we think are less principled and can cause problems when reactive and flexible controls are needed.
> > >
> > > > My point is that we cannot say that the action space of Go is much smaller than the continuous control problems just because the stone of Go is put on a 2-dimensional board. The action space of Go is naturally discrete and the difficulty of the game depends on the number of grids on the board.
> > >
> > > If converting discrete action spaces into continuous ones doesn't make sense to you, we can also consider converting continuous action spaces back to discrete ones. TT discretization will break each continuous dimension into 100 buckets and the size of the discrete action space is thus $100^D$, where $D=24$ for Adroit robotic hand control. This number is much larger than $19^2$ of the game of Go.
> > >
> > > > In Go, it is obvious that the Values of different positions are significantly different (i.e., the curve of the value function changes drastically), but the curves of the action-value functions in continuous control tasks are smooth, continuous, and changing slowly.
> > >
> > > We agree with your claim here intuitively. On the other hand, we think this is orthogonal to the size of the action space but more about the decision-making of the game of Go is more complex than controlling a hopper to move forward. If we consider a complex game with continuous control, like playing football, a minor difference in motor control can lead to a drastic change in the value.

---

### Official Review · Reviewer_rxeb · 2022-10-25

**Confidence:** 3
**Correctness:** 3
**Technical Novelty And Significance:** 4
**Empirical Novelty And Significance:** 3
**Recommendation:** 8

**Clarity, Quality, Novelty And Reproducibility:**

The paper is mostly well written, novel and appears reproducible.

Some things that could be improved:
- Framing: The intro implies planning-based RL has only shown strong performance on low-dim problems, but as e.g. Go, Chess are quite high-dimensional, this claim seems questionable. I would consider rephrasing that, I think the paper can make its point about improved efficiency in high-dim planning without it.

- How much data was actually used to train the offline model in the experiments? Has to be quite a bit?

- The paper is simultaneously also talking about RL success and compute, but unless the model is actually better (and not just more compact) RL success for earlier approaches should also depend on the compute spent on sampling the (less compact) space? I feel these aspects could have been better examined separately, but I realize space is at a premium.

Some minor language issues:
- "future trajectories estimated by a dynamics model" -> sampled from?
- The second paragraph of the intro is a bit wordy, and many arguments seem to overlap. I think it could be condensed a bit.
- I didn't see the VQ-VAE paper cited. The explanation of VQ-VAE was also terse to the point of being mysterious.
- Typo on p.4: "probabiilty"


**Strength And Weaknesses:**

Strengths:
- The idea of learning a latent discrete action space is very interesting and highly relevant
- The approach appears sound and novel
- The paper is well-written with a thorough ablation study

Weaknesses:
- The experiment tasks, while common in this type of paper, are rather simple. The locomotion problems are not complex enough to show improvement, and for the robotic hand experiments, the optmimal length L of the latent trajectory segments seems rather small (3). This makes it a bit unclear how much the learned latent representation helps with planning compared to just 1) reducing the time discretization of the TT by a factor of 1/L (i.e. executing the same action for L=3 steps in a row), and 2) manually discretizing the action space (e.g. clustering it using k-means), although for a fair comparison one would need to sample from a learned conditional sequence prior (or ignore it for both) 3) both of these in combination. I do not think this was covered by the existing baselines? Would 1) be feasible to add?
- It would have been interesting with some introspection and qualitative examples of the learned latent action representation (trajectory segments). This approach should be more suitable for tasks that have a natural discrete structure. It would have been interesting to see how well it could recover that.


**Summary Of The Paper:**

The authors extend the Trajectory Transformer (TT) by learning a latent discrete action-space consisting of trajectory segments, which is used to speed up (beam-search) planning for offline RL. The authors argue that planning in this latent space scales better with dimensionality of the action space. Additionally, the proposed architecture is computationally cheaper than the TT.

**Summary Of The Review:**

This seems like a good paper, presenting novel and relevant ideas for learning a discrete action space in offline RL. I think their analysis of the learned representations (and in particular how much model accuracy vs. compactness contributes the improved success rate) could have gone a bit further, but it does appear sound, results in improved or similar results compared to earlier work and has a large ablation study already.

EDIT: I just noticed that some seemingly relevant but unlisted related work surfaced in the other reviews. I will adjust my score based on the outcome of that discussion.

---

> ### Author Response · Authors · 2022-11-14
> **Individual Response**
>
> Many thanks for your strong support of our work and your constructive feedback!
>
> > EDIT: I just noticed that some seemingly relevant but unlisted related work surfaced in the other reviews. I will adjust my score based on the outcome of that discussion.
>
> We’ve updated the paper to include all the related mentioned by other reviewers and we believe they do not affect the novelty and originality of our work. Please see the general response for a summary of the novelty and differences to related work, and check out individual responses if you want to dive into details.
>
> > This makes it a bit unclear how much the learned latent representation helps with planning compared to just 1) reducing the time discretization of the TT by a factor of 1/L (i.e. executing the same action for L=3 steps in a row)
>
> Since TT (and similarly Gato) discretize each state/action dimension separately and treats each dimension as a token, even if we reduce the time resolution of TT by a factor of ⅓, the overall sequence length of TT is still D times longer than the TAP (D can as large as 200 for adroit). We would expect such a modification to only marginally improve the planning efficiency while damaging the performance of TT.
>
> > 2) manually discretizing the action space (e.g. clustering it using k-means), although for a fair comparison one would need to sample from a learned conditional sequence prior (or ignore it for both)
>
> The discretization of TT (and Gato) also leverages the statistical information of the whole dataset, which can be regarded as per-dimensional clustering. K-means clustering over the whole action space might be an interesting alternative over TT and Gato discretization but it will still be quite different from the latent action space of TAP. Basically, the latent action of TAP can only be decoded back to actions conditioned on the current state. It will be better to think TAP did clustering in the space of policy rather than the raw action space. The marginal distribution of raw actions can have much more modes than the policy (conditional distribution of actions) therefore it will probably need quite a lot of discrete buckets to accurately describe them, and we expect it will be similar to TT discretization rather than TAP.
>
> > The locomotion problems are not complex enough to show improvement, and for the robotic hand experiments, the optmimal length L of the latent trajectory segments seems rather small (3)
>
> Now to directly respond to the critique, we agree that the locomotion environments are not complex enough to show the performance improvement of TAP over TT. But the planning time advantage is still very obvious because the time complexity of TT planning is $O(D^3T^3)$ but that of TAP is $O(L^{-1}T^3)$, where D is the dimensionality of the state plus that of the action.
> And for $L$ being small, as you might have noticed, we did ablation over $L$ over locomotion control (Figure 3 and Table 7 in the appendix) and it does show $L>1$ is helpful for the performance. We think this shows the contribution of temporal abstracted action space is a valid contribution. For adroit, we didn’t deliberately try to optimize $L$ because we don’t think $L$ is not the main contribution of the difference between TT and TAP but we can add such ablation if you think that will be helpful.
>
> > It would have been interesting with some introspection and qualitative examples of the learned latent action representation (trajectory segments).
>
> Thanks for the advice! We include visualization of predicted trajectories and latent codes in the updated version, Appendix K Figure 8. Actually, the three trajectories in Figure 2 are also trajectories generated by a TAP agent trained on hopper-medium-replay.
>
>
> > The intro implies planning-based RL has only shown strong performance on low-dim problems, but as e.g. Go, Chess are quite high-dimensional, this claim seems questionable. I would consider rephrasing that, I think the paper can make its point about improved efficiency in high-dim planning without it.
>
> For low-dim problems, we actually mean low action dimensionality. For every single step, the stone of Go can only be put on the 2-dimensional board and its action can be easily described by a tuple of two integers (x,y). For Chess a move can be described by starting positions and ending positions, therefore 4-dimensional. To be precise, when we consider an action space to be N-dimensional if it has N degrees of freedom, irrelevant to how many discrete buckets there are. On the other hand, if we consider the number of discrete buckets, Go has $19^2$, TT discretization over Adroit gives $100^{24}$ buckets for every action (that of TAP discretization of Adroit is $512^{1/L}$). Therefore, the point is, measuring the number of discrete buckets or measuring degrees of freedom will give the same conclusion that the action space of Go and Chess is much smaller than high-dimensional continuous control.

---

> > ### Author Response · Authors · 2022-11-14
> > **Individual Response (2)**
> >
> > > How much data was actually used to train the offline model in the experiments? Has to be quite a bit?
> >
> > Since we are doing experiments on the D4RL dataset, the overall number of transitions are less than 1M. For adroit-human, the number of transitions can be as low as 5000. None of them can be regarded as very large data. However, it’s definitely an exciting research direction to train TAP on larger datasets, which might further unleash the power of these generative models that have shown strong performance on CV/NLP.
> >
> > > The paper is simultaneously also talking about RL success and compute, but unless the model is actually better (and not just more compact) RL success for earlier approaches should also depend on the compute spent on sampling the (less compact) space? I feel these aspects could have been better examined separately, but I realize space is at a premium.
> >
> > This is a very good point. In this paper, we did not investigate if the TAP model is more accurate than TT and our focus is the difficulty of test-time planning (optimization). For TT on adroit, we tried to keep the planning hyper-parameters the same as that on gym-locomotion but it’s already quite expensive (30 seconds for a single step of decision). One can argue that further improving the planning budget can improve the performance of TT but it’s unclear how much more computation is really needed. If that’s going to be something like 1 hour, then the practical value of having such a model might be low, even if it can give the same final performance as TAP.
> > On the other hand, we are rather confident that the model accuracy of TAP within the action support is more accurate than the traditional one-step model used by MOPO. As in adroit experiments, Opt-MOPO did per-task bayesian optimization of hyper-parameters while its final performance is still surpassed by TAP by a large margin and also beaten by TT.
> > We believe a rather conservative claim is TAP extends the strong modelling accuracy of TT (because using a strong generative model) while making planning much easier and therefore becoming a practical algorithm for high-dimensional control.
> > While this is the main point we want to sell in the previous paper, we agree having more study on compactness v.s accuracy is a good direction to go for the next iteration of the work.

---

### Author Response · Authors · 2022-11-14
**General Response**

## Summary of Positive Feedback
We thank the detailed and constructive feedback from all the reviewers. It appears the reviewers appreciate different aspects of our work, which we summarise here:
* Reviewer rxeb likes the idea of learning and planning in discrete latent action space with temporal abstraction.
* Reviewer Brj7 appreciates both spatial and temporal abstraction which reduces the computational complexity of TT, a strong model-based method that has strong prediction accuracy but suffers from high computational cost, especially for planning.
* Reviewer Ykqw likes the high-level idea of learning a dynamics model defined in a latent action space which restricts search space and avoids querying out-of-distribution actions.

There are also pros that multiple reviewers agree on:
* All the reviewers agree the writing is clear and easy to follow
* Reviewer rxeb and Ykqw like the comprehensive ablation study

## Novelty and related work

We thank reviewers Ykqw and Brj7, and rxeb for pointers to related work. We’ve updated the related work to include the works in the domain of model-free RL with latent actions, goal-oriented imitation learning, and teleportation with a learned action space. All the updates are marked as green.
Here we summarise our novelty and a more detailed comparison with each small class of works can be found to the individual responses.

### Short version

**1. Methodology**: TAP learns a state-conditional trajectory generative model with VQ-VAE and does multi-step planning in the bottleneck of the VQ-VAE. Previous related methods (mentioned by the reviewers) do not touch that and lead to qualitatively different latent actions.

**2. Conceptually**: This paper proposes a solution to address planning in a high-dimensional continuous action space, that is to plan in a learned low-dimensional action space. Previous work motivates latent actions from the perspective of an action constraint in a model-free/imitation setting and is not relevant to multi-step planning.

**3. Empirically**: We managed to achieve state-of-the-art performance on D4RL Adroit within the model-based family, which used to be very difficult for previous model-based methods because of the high action dimensionality.


### Longer Clarification on the methodology differences,
1) TAP is a model-based method where the decoder acts as a predictive model which is the central topic of this work. The point of having a latent action space is for efficient planning with the predictive model, so the main contribution of this work is learning a latent action model and efficient planning. Note of the related work operates in a model-based, planning setting. [1][4][5] are model-free; [2][3] did goal-oriented imitation; [6][7] provides an interface for human control.

2) The word latent action is quite a broad concept. TAP latent actions are not a representation of raw actions because they cannot be reconstructed without the current state. In addition, the latent actions of TAP are a structured sequence of latent codes where the first $N$ latent codes can be partially decoded into $NL$ steps of transitions and the meaning (raw actions being decoded) of later latent codes are conditioned on the earlier ones. These features enable efficient multi-step planning of TAP. For [1][4][5] there are no temporal abstractions. [3][4] are only vaguely related to latent actions (see the response to reviewer 3 for details). For [2][5][6][7] the latent action space is not discrete. For [1][2][3][4][5][6][7] and others we mentioned in the related work, the latent actions don’t have sequence structures.
3) TAP allows decoupling between decision (planning) steps and the actual environment steps in a flexible way. In our main text, we group a fixed number of $L$ consecutive actions, which is just one example. In Appendix D, as **updated in Table 10 in the revision**, we showed the coupling between latent planning steps and decoded trajectory steps can be learned with a cross-attention without hand-crafted segments. Such a latent action space gives an even better performance when applying vanilla sampling, which shows the further potential of TAP.

[1] Continuous Control with Action Quantization from Demonstrations

[2] IRIS: Implicit Reinforcement without Interaction at Scale for Learning Control from Offline Robot Manipulation Data

[3] Learning latent plans from play data, Lynch et al., 2019

[4] LASER: Learning a Latent Action Space for Efficient Reinforcement Learning, Allshire and Martin-Martin et al, 2021

[5] PLAS: Latent Action Space for Offline Reinforcement Learning, Zhou et al., 2020

[6] Learning latent actions to control assistive robots, Losey et al., 2022

[7] Learning visually guided latent actions for assistive teleoperation, Karamcheti et al., 2021

---

### Decision · Program_Chairs · 2023-01-20

**Decision:**

Accept: poster

**Justification For Why Not Higher Score:**

The paper builds on well known ideas and is addressing a somewhat narrow problem.

**Justification For Why Not Lower Score:**

A solid paper, the most negative reviewer did not react but I think the concerns have been well addressed.

**Metareview: Summary, Strengths And Weaknesses:**

Summary:
The paper extends trajectory transformers with a latent action encoding for model-based / planning-based RL. The approach is evaluated in two benchmark suits.

Strengths:
- Interesting approach that works well
- Well written paper
- Most concerns of the reviewers have been successfully addressed in the discussion (also of the one that didn't reply)

Weaknesses:
- The locomotion problems add only limited insights
- The general idea of action abstractions is well known, this particular setting seems new

**Note From Pc:**

if the above contains the word "oral" or "spotlight" please see: "oral" presentation means -> notable-top-5% and "spotlight" means -> notable-top-25%. As stated in our emails, we are disassociating presentation type from AC recommendations

**Summary Of Ac-Reviewer Meeting:**

N/A